# Cerebellar growth is associated with domain-specific cerebral maturation and socio-linguistic behavior

Aikaterina Manoli [1,2,3] ✉, Neville Magielse [1,2,4], Felix Hoffstaedter [2,4], Nilsu Sağlam[1], Thanos Tsigaras [2,4], Augustijn A. A. de Boer[5,6], Lorenz Ahle[1], Ceyda Yalçin[1], Milin Kim [7,8], Torgeir Moberget [7,9], Thomas Wolfers [7,10,11], Casey Paquola [2], Charlotte Grosse Wiesmann [1,12], Andre F. Marquand[5,6,13], Jorn Diedrichsen [14,15,16] & Sofie L. Valk [1,2,4] ✉

The cerebellum's involvement in cognitive functions is increasingly recognized, yet its developmental contribution to cognition remains poorly understood. The cerebellum undergoes rapid development in early life, paralleling major cognitive and behavioral changes. Although clinical studies have linked early cerebellar disruptions to profound developmental deficits, it remains largely unclear how typical cerebellar maturation supports the development of cognitive functions and how it interacts with broader cerebral development. Here, we apply a normative modeling framework to map cerebellar volumetric growth from age one to young adulthood ($N = 751$; ages 1–21 years). Using both lobular and functional cerebellar parcellations, we characterize typical cerebellar development from late infancy and its relationship to cerebral development and behavioral performance in childhood through adulthood. Across parcellations, association areas consistently show steeper growth trajectories than sensorimotor areas. Cerebellar and cerebral areas with similar functional roles demonstrate coordinated maturation, and volumetric growth in the posterior cerebellum relates to individual differences in socio-linguistic behaviors. These findings establish a comprehensive reference for typical cerebellar development, highlight cerebellar co-maturation with the cerebral cortex, and underscore the cerebellum's role in supporting the development of cognitive functions.

The cerebellum's vast computational architecture is increasingly recognized as a "universal learning system"[1–3] that supports both sensorimotor and cognitive processes, including language, memory, and social cognition[4–7]. The expansive functional cerebellar repertoire is suggested to be largely mediated through extensive reciprocal connections with the cerebral cortex[8–11]. Current theoretical frameworks propose that the cerebellum encodes internal models of actions and behaviors that are transmitted to cerebral regions to facilitate automated behavioral control and optimization[1,3,12,13]. Despite substantial evidence supporting the cerebellum's role in diverse cognitive processes, its specific role in cognitive development remains poorly characterized.

The cerebellum exhibits a protracted developmental trajectory, emerging as one of the first brain structures at approximately thirty days post-conception and being among the last to achieve maturation in the first postnatal years[14–16]. This extended developmental timeline

confers particular vulnerability to environmental perturbations that may disrupt normal cerebellar maturation. Lesions in early life, when the cerebellum is still rapidly developing, result in profound and persistent cognitive impairments, which have disproportionate effects on socio-linguistic functions compared to lesions sustained in adulthood[17,18]. Moreover, cerebellar abnormalities, including volumetric reductions and atypical functional activation patterns, are consistently associated with neurodevelopmental conditions such as autism, attention deficit hyperactivity disorder (ADHD), and schizophrenia[19–21], as well as deficits across multiple cognitive and affective domains[22]. Characterizing typical cerebellar development is therefore essential for identifying critical periods during which developmental disruptions may exert substantial and enduring effects on neurodevelopmental and cognitive outcomes[23].

Normative modeling offers a powerful framework for characterizing cerebellar developmental trajectories. By modeling developmental trajectories across many individuals, population-based normative models provide a reference that captures the expected range of typical development within a neural circuit[24,25]. Unlike traditional group-average approaches that assume homogeneity within populations, normative modeling quantifies individual deviations from typical patterns, enabling the identification of variable developmental profiles and their associations with cognitive and behavioral performance[26,27]. Furthermore, this framework supports robust modeling of normative development using parameters derived from large, multi-site datasets and can be flexibly updated as new data become available. Currently, a unified normative model spanning the entire cerebellar developmental trajectory from infancy through young adulthood is lacking. Recent normative modeling studies focusing on childhood and adolescence have revealed that motor and association cerebellar subregions mature at different rates[28,29]. These findings are complemented by longitudinal research demonstrating rapid growth across all cerebellar lobules in the first two years of life, and especially during the first seven months after birth[30]. Despite this progress, a comprehensive reference model spanning infancy to adulthood remains critically needed, given the different rates in which the cerebellum matures in the first postnatal years compared to childhood and adolescence[14–16,18]. Addressing this gap is important, as cerebellar subregions may exhibit divergent maturation patterns across developmental stages, potentially indicating periods when different regions are most susceptible to developmental perturbations[21]. Establishing a comprehensive reference model of cerebellar growth from the first years of life to adulthood would thus enable more precise estimates of individual variability related to both typical and atypical cerebellar development.

Furthermore, a critical aspect of understanding cerebellar regional development involves examining its relationship with cerebral cortical maturation. The cerebellum does not act as an isolated neural circuit but interacts extensively with the cerebral cortex through numerous reciprocal pathways[31], where it plays a fundamental role in supporting cerebral functions[1,13,32,33]. It has been proposed that cerebellar topography may largely mirror that of the cerebral cortex[34]. Consequently, functionally related regions of the cerebellum and cerebral cortex may exhibit coordinated maturation, with areas subserving similar functions displaying similar growth patterns[35]. By mapping typical cerebellar and cerebral development together, we can better understand how their coordinated growth supports integrated brain function and how variation in one system may be associated with alterations in the other.

Finally, understanding how cerebellar development contributes to cognitive and behavioral capabilities remains a key area for investigation. Functional magnetic resonance imaging (fMRI) studies[36–38] and meta-analytic investigations[39,40] have established detailed functional parcellations of the adult cerebellum based on its involvement in diverse motor, affective, and association domains. These studies reveal that anterior cerebellar regions are primarily involved in sensorimotor functions, while posterior regions, particularly Crus I and II, are engaged in associative functions, including language and social cognition[4,6]. However, how the cerebellum supports these functions during development remains poorly understood. Posterior cerebellar maturation is linked to early-life acquisition of several behaviors, including fine motor skills[30] and social cognitive abilities[32], as well as vulnerability to neurodevelopmental disorders[28,29]. This diversity of behavioral performance highlights the need for a comprehensive and detailed investigation of cerebellar growth trajectories and their relation to behavioral development.

To investigate cerebellar development and how it relates to that of the cerebral cortex and behavioral performance, the present study describes normative models of cerebellar volumetric growth from late infancy to young adulthood (1–21 years). We used cross-sectional infant data (1–5 years; $N = 111$) from the Baby Connectome Project (BCP)[41] and child and adolescent data (5–21 years; $N = 640$) from the Lifespan Human Connectome Project in Development (HCP-D)[42]. Recognizing that functional boundaries in the cerebellum do not coincide well with the anatomical, lobular boundaries[4], we examined cerebellar development across both lobular and functional parcellations to comprehensively characterize subregional development in relation to diverse cognitive functions. For lobular boundaries, we applied a deep learning segmentation of cerebellar lobules (ACAPULCO)[43]. For functional boundaries, we leveraged multiple atlases: cerebellar resting-state networks[36], a parcellation based on a multi-domain task battery (MDTB)[4], and a parcellation constructed by fusing diverse task-based datasets into a single functional atlas[6]. We further associated cerebellar normative growth trajectories with developmental patterns of the cerebral cortex to examine maturational coupling between cerebellar and cerebral subregions. Finally, we investigated how subregional cerebellar growth relates to performance across a range of behavioral domains, including sensorimotor abilities, language, memory, and social cognition. Our results show that posterior cerebellar regions involved in associative processes undergo more rapid developmental growth than sensorimotor regions from approximately one year of age through adulthood. We also find evidence of coordinated maturation between cerebellar and cerebral regions with similar functional roles. Notably, we observe a selective relationship between posterior cerebellar maturation and individual variation in socio-linguistic abilities. This is particularly strong for social behavior, where cerebellar development explains variance beyond what cerebral cortical development alone can account for. Together, these findings deepen understanding of how cerebellar development aligns with cerebral maturation and supports emerging cognitive abilities, with potential relevance for identifying biomarkers of atypical neurodevelopment.

## Results

### Posterior cerebellar lobules exhibit steeper trajectories than anterior lobules from late infancy to adulthood

We generated normative models of cerebellar anatomical lobules for ages 1–21 with Hierarchical Bayesian Regression (HBR) implemented in the PCNtoolkit[25,44,45]. Both linear and third-order B-spline models were generated. Both performed equally well based on leave-one-out cross-validation (LOOCV; Supplementary Table 5). For parsimony, we here described linear models, though B-spline models are also provided as they may offer greater flexibility for future applications[28]. Posterior distributions for all model parameters and parcels demonstrated good convergence (over 95% of R-hat values below 1.01[46]).

The lobular segmentation underwent rigorous quality control, with each image independently assessed and, if necessary, carefully manually corrected by multiple reviewers through a consensus-based procedure (see "Methods": Cerebellar parcellation). Lobules demonstrated volumetric increases from late infancy to adulthood (Fig. 1,

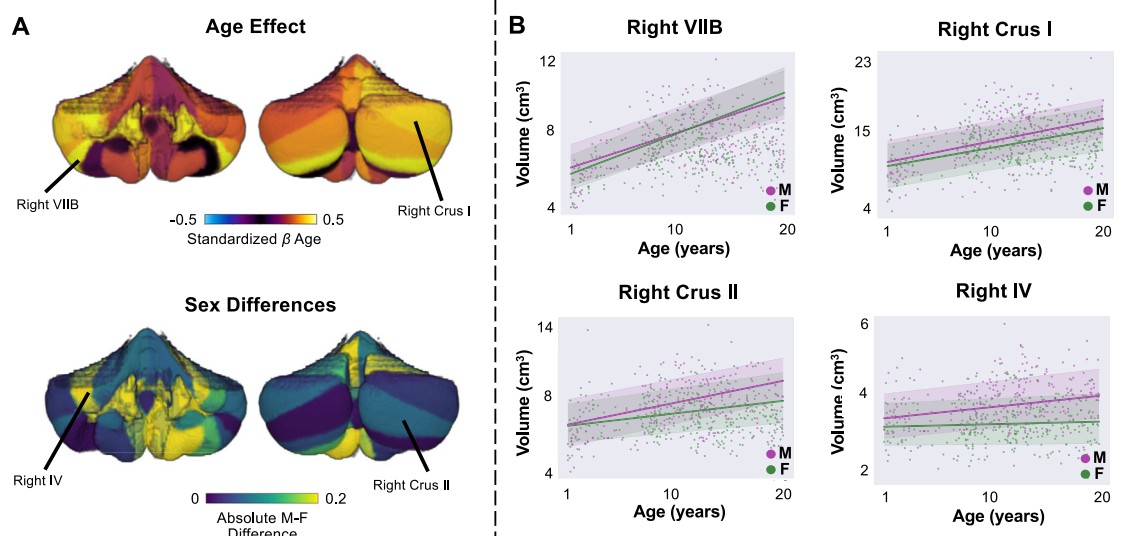

**Fig. 1 | Normative growth trajectories for cerebellar anatomical lobules from late infancy to adulthood (1–21 years). A** Mean effect of age on growth of anatomical lobules (top) and absolute male-female sex differences (bottom). Mean $\beta_{age}$ values for anterior lobules I-IV are combined on the cerebellar plot. **B** Sex-stratified normative trajectories of four of the lobules with the steepest developmental trajectories. Bold lines represent the mean trajectories per sex. Shaded areas represent the 68% CI. Trajectories are shown on parcel-specific y-axis scales to preserve differences in absolute volume and to visualize growth relative to each lobule's baseline size. Source data are provided as a Source Data file. M male, F female.

Supplementary Fig. 5, and Supplementary Table 1), with the steepest trajectories observed in the bilateral VIIB (left: $\beta_{age} = 0.63$; 95% CI = [0.18 1.52]; right: $\beta_{age} = 0.79$; 95% CI = [0.23 1.73]), followed by the right VI ($\beta_{age} = 0.47$; 95% CI = [-0.16 1.44]), right Crus I ($\beta_{age} = 0.47$; 95% CI = [−0.14 1.06]), right IV ($\beta_{age} = 0.36$; 95% CI = [−0.14 0.81]), and the bilateral Crus II (left: $\beta_{age} = 0.36$; 95% CI = [−0.21 1.18]; right: $\beta_{age} = 0.37$; 95% CI = [−0.17 1.17]). Overall, posterior lobules (VI-X: overall mean $\beta_{age} = 0.29$) exhibited steeper trajectories than anterior lobules (I-V: overall mean $\beta_{age} = 0.25$). A notable exception was the bilateral posterior VIIIA, primarily involved in motor processing[37,39], which demonstrated flatter trajectories compared to other posterior lobules (left: $\beta_{age} = 0.01$; 95% CI = [−0.48 0.48]; right: $\beta_{age} = 0.05$; 95% CI = [−0.50 0.61]). We also observed subtle sex differences across parcellations, with age-related coefficients being slightly higher for males ($\beta_{age}$ across parcels = 0.29) than for females ($\beta_{age}$ across parcels = 0.28). The largest sex differences were observed in the left IX (mean absolute difference in $\beta_{age} = 0.38$) and left V (mean absolute difference in $\beta_{age} = 0.34$; Fig. 1A, bottom).

**Cerebellar association regions exhibit steeper growth trajectories than sensorimotor regions from childhood to adulthood**
Next, we generated normative models for three widely used functional parcellations derived from task-based and resting-state neuroimaging data[4,6,36] to comprehensively describe normative cerebellar development across different functional taxonomies. The parcellations and associated parcel labels can be seen in detail in Supplementary Fig. 4. Consistent with the lobular analyses, we describe results from linear models, which showed predictive performance similar to that of third-order B-spline models as assessed by LOOCV (Supplementary Tables 6-8).

We restricted analyses to participants aged 5 years and older, as functional spatial patterns in the first years of life may differ from those in adults but show similarity with adult functions later in childhood[47–49]. General normative developmental patterns mirrored those of the lobular trajectories and were consistent across parcellations. Specifically, we again observed age-related volumetric increases across all functional parcellations in both males and females. In all parcellations, sensorimotor parcels, primarily localized

in the anterior cerebellum, exhibited smaller age-related effects and less steep growth trajectories compared to parcels related to a range of association processes (e.g., language, memory; Fig. 2A-C), primarily localized in the posterior cerebellum. For instance, left S1 of the fusion atlas (linguistic processing) showed the highest age effect ($\beta_{age} = 0.42$; 95% CI = [0.32 0.52]), while left M4 (lower limb motor processing) showed the lowest ($\beta_{age} = 0.08$; 95% CI = [−0.01 0.32]). Despite overall consistency in flatter growth trajectories for sensorimotor regions and steeper trajectories for association regions, we observed modest differences across corresponding areas between parcellation schemes. In the fusion and MDTB atlases, left-lateralized association regions−such as D1 (spatial working memory) and S1 (linguistic processing) in the fusion atlas, and regions 5 (divided attention) and 7 (narrative processing) in the MDTB atlas−showed stronger age-related effects. In contrast, in the resting-state atlas, the corresponding dorsal attention and default mode networks exhibited age effects that were more bilaterally distributed across hemispheres.

Across all parcellations, we also observed sex differences, with males exhibiting overall higher age-related coefficients than females (Table 1). These differences were most pronounced in the right D2 (retrieval; mean absolute difference in $\beta_{age} = 0.20$), left M3 (hand and upper body; mean absolute difference in $\beta_{age} = 0.20$), and right S2 (social processing; mean absolute difference in $\beta_{age} = 0.20$) in the fusion atlas and region 1 (left hand press) in the MDTB atlas (mean absolute difference in $\beta_{age} = 0.20$). Mean age effects and normative trajectories of all functional parcels are provided in Supplementary Figs. 6−8 and Supplementary Tables 2-4.

**Cerebellar and cerebral subregions demonstrate domain-specific maturational coupling**
Next, we examined cerebellar and cerebral co-maturation patterns based on normative parcel trajectories. Using Lasso regularized regression, we predicted z-scores in the Yeo et al. 17-network atlas[50] from z-scores of lobular and functional cerebellar parcels (see "Methods": "Association with normative trajectories of the cerebral cortex"). Given the consistency observed among functional parcellations, we only focused on z-scores derived from the functional fusion atlas (5−21

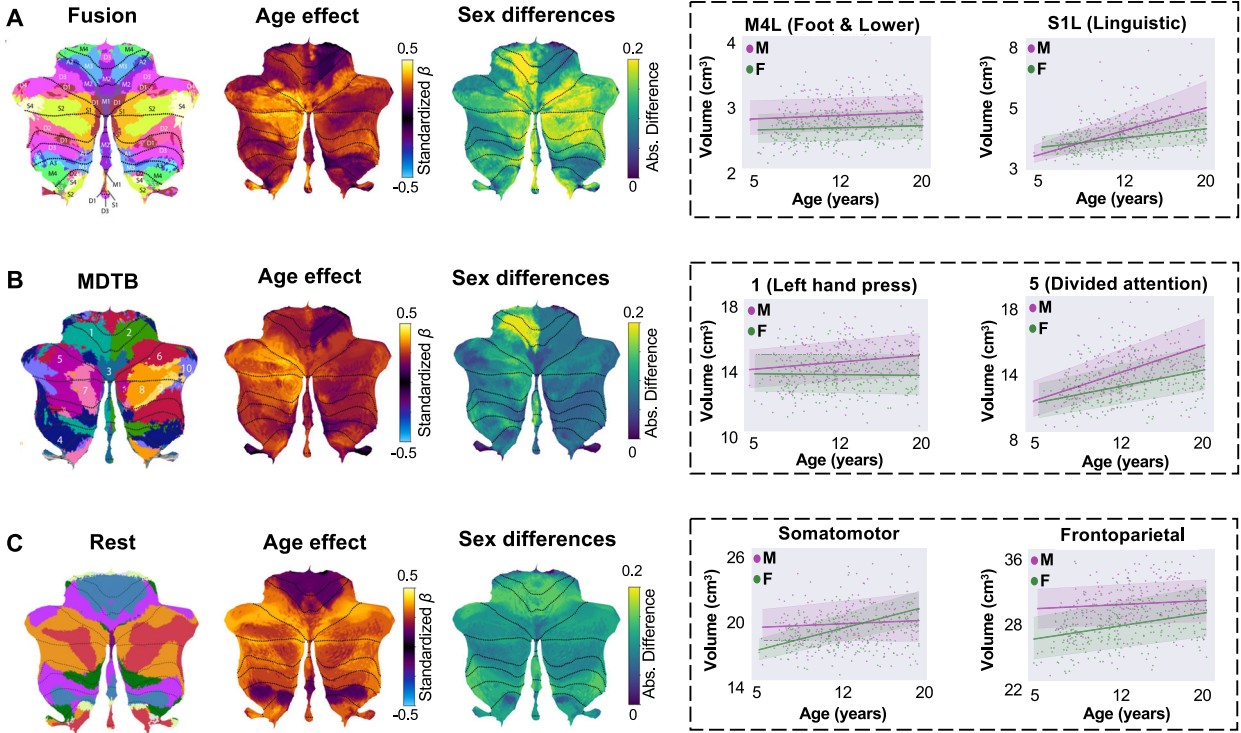

**Fig. 2 | Normative growth trajectories for cerebellar functional parcels from childhood to adulthood (5-21 years). A–C** Mean age effect, sex differences and example trajectories for sensorimotor and association parcels are shown for the fusion (**A**), MDTB (**B**), and resting-state atlas (**C**). Left: Atlas images, adapted from Nettekoven et al. (fusion)[6], King et al. (MDTB)[4], and Buckner et al. (rest)[36]. Middle: Mean effect of age and absolute male-female sex differences of functional lobules.

Right: Sex-stratified normative trajectories of example sensorimotor and association parcels. Bold lines represent the mean trajectories per sex. Shaded areas represent the 68% CI. Trajectories are shown on parcel-specific y-axis scales to preserve differences in absolute volume and to visualize growth relative to each lobule's baseline size. Source data are provided as a Source Data file. MDTB multi-domain task battery, Abs. absolute, M male, F female, L left, R right.

**Table 1 | Sex-stratified means of standardized $\beta_{age}$ across functional parcels**

| Atlas | Male (Mean [95% CI]) | Female (Mean [95% CI]) |
|---|---|---|
| Fusion | 0.31 [0.19 0.45] | 0.17 [0.06 0.29] |
| MDTB | 0.31 [0.19 0.43] | 0.21 [0.12 0.32] |
| Rest | 0.28 [0.22 0.46] | 0.25 [0.14 0.34] |

years; Fig. 3A), as a more comprehensive taxonomy of cerebellar functions resulting from multiple large-scale datasets[6].

Cerebral parcels in the 17-network atlas consistently exhibited volumetric decrease, with most pronounced effects observed in the left-hemisphere network 17, corresponding to the default mode network ($\beta_{age}$ = -0.49; 95% CI = [-0.57 -0.40]; Fig. 3B and Supplementary Fig. 10). Cerebellar and cerebral parcels that subserved similar cognitive functions were significantly associated at the level of individual cerebellar-cerebral coefficients [10,000 permutations; one-sided and corrected for the false discovery rate (FDR)[51] at $q = 0.05$], revealing domain-specific maturational coupling. The most prominent associations within functional domains of the fusion atlas[6] (i.e., socio-linguistic, multi-demand, motor) exemplify this domain-specific pattern. In the socio-linguistic domain, right S2 (corresponding to social processing) and S3 (corresponding to resting-state activation) were primarily associated with default-mode network regions (e.g., networks 16-17). In the multi-demand domain, right D2 (corresponding to working memory), was associated with attention, memory, and executive control regions (e.g., dorsal attention network 5, salience networks 7-8, and limbic network 9). Right M2 (corresponding to mouth movements) in the motor domain was strongly correlated with

sensorimotor networks (e.g., visual network 2 and somatomotor network 4). Lastly, right M3 (corresponding to lower limb movements) was strongly anticorrelated with default-mode networks 15–17, but notably also with somato-motor network 3. No significant associations were found for the action domain at the FDR threshold (Fig. 3D).

Importantly, these domain-specific patterns remained largely consistent when associating cerebellar trajectories with a gyral-based parcellation of the cerebral cortex [Desikan-Kiliany (DK) atlas[52]], suggesting that the observed cerebello–cerebral coupling captures stable, biologically meaningful functional relationships that extend beyond a single cerebral parcellation scheme. Specifically, in the socio-linguistic domain, right S3 (corresponding to resting-state activation) was primarily associated with default-mode network regions (e.g., precuneus[53], posterior cingulate[54]). In the multi-demand domain, right D2 (corresponding to working memory), was associated with memory and executive control regions (e.g., entorhinal[55], caudal middle frontal gyrus[56]). Right M4 (corresponding to lower limb movements) in the motor domain was anticorrelated with several associative cerebral regions (e.g., precuneus[53], supramarginal gyrus[57]). Lastly, left A1 (corresponding to spatial simulation, such as mental imagery and navigation) in the action domain was linked to sensory and perceptual regions (e.g., paracentral[55], postcentral gyrus[58,59];Supplementary Fig. 11; see Supplementary Fig. 12 for hemisphere-specific associations in the heatmap).

In addition to associations with the functional fusion atlas, we also examined the relationship between lobular cerebellar trajectories and cerebral development. Here, we focused on comparisons with the DK atlas to enable a direct anatomy-to-anatomy comparison. This revealed similar maturational coupling patterns, particularly in posterior lobules, whose growth trajectories were related to cerebral association

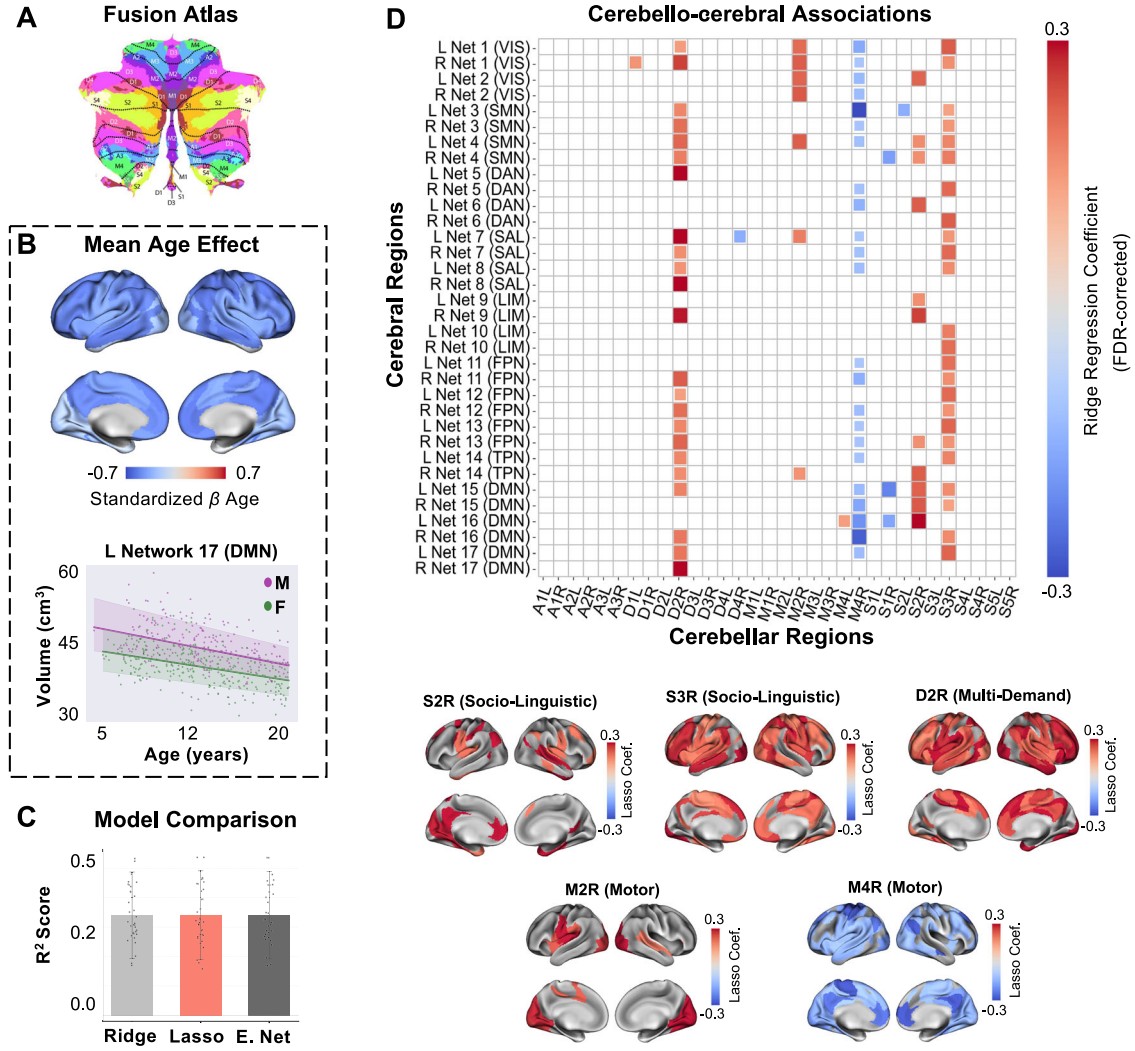

**Fig. 3 | Association of cerebellar (fusion atlas) and cerebral (Yeo et al. 17-network atlas) growth trajectories. A** Functional fusion atlas (adapted from Nettekoven et al.[6]). **B** Mean effect of age on the Yeo et al. 17-network atlas (top) and example normative trajectory for right network 17 (default mode; bottom). Bold lines represent the mean trajectories for each sex. Shaded areas represent the 68% confidence interval. **C** Comparison of Ridge, Lasso, and ElasticNet regularization models based on average $R^2$ scores (10-fold cross-validation). The Lasso model marginally outperformed the other two and was selected (mean $R^2$ = .34). Data points denote parcel-level cross-validation performance. Error bars represent the standard error of the mean. **D** Top: Significant cerebello-cerebral associations (10,000 permutations, FDR-corrected at $q$ = 0.05). A global $\alpha$ = 0.1 was selected for associations based on a multi-output Lasso model predicting all cerebral parcels simultaneously and aligns with the distribution of parcel-wise optimal $\alpha$ values, where over 80% of cerebral parcels individually favored $\alpha$ = 0.1. Bottom: FDR-corrected weights for cerebellar parcels with the largest number of significant associations in the fusion atlas, projected on the Yeo et al. 17-network atlas. Source data are provided as a Source Data file. Coef. coefficient, E. Net ElasticNet, VIS visual network, SMN somatomotor network, DAN dorsal attention network, SAL salience network, LIM limbic network, FPN frontoparietal network, TPN temporoparietal network, DMN default mode network, M male, F female, L left, R right.

regions. However, these findings demonstrated less domain specificity, which could be explained by the observation that multiple functional processes map to each cerebellar lobule[4] (Supplementary Fig. 13). Overall, cerebello-cerebral maturational coupling followed functional network organization, with functionally related regions exhibiting coordinated developmental trajectories across brain structures.

## Growth of cerebellar association regions corresponds to socio-linguistic behavioral development

Lastly, we used Partial Least Squares (PLS) analysis to link individual deviations in normative cerebellar development with individual differences in a broad range of behavioral markers. PLS identifies latent variables that capture shared patterns of variance between parcel-level $z$-scores and behavioral performance[60]. We applied this approach to examine associations between normative $z$-scores across fusion atlas parcels (ages 5-21; HCP-D dataset) and performance on sensorimotor and association tasks (see "Methods": "Behavioral tasks" and Supplementary Fig. 14 for task score distributions). Behavioral scores were standardized and residualized for age and sex to be consistent with the parcel $z$-scores. PLS analysis extracted a significant latent variable relating parcel growth and behavioral scores ($p$ = 0.010, one-sided). This single latent variable accounted for 78% of the covariance across the two variables (Fig. 4A; top). Next, the PLS model was evaluated using 10-fold cross-validation (Fig. 4A; bottom). The mean training set correlation across folds was $r(113)$ = 0.31, and the mean test set correlation was $r(11)$ = 0.27. The empirical correlation between behavioral scores and parcel $z$-scores was $r(128)$ = 0.30, $p$ = 0.001 (one-sided and tested against a null model assuming no brain-behavior relationships over 10,000 permutations; Fig. 4B).

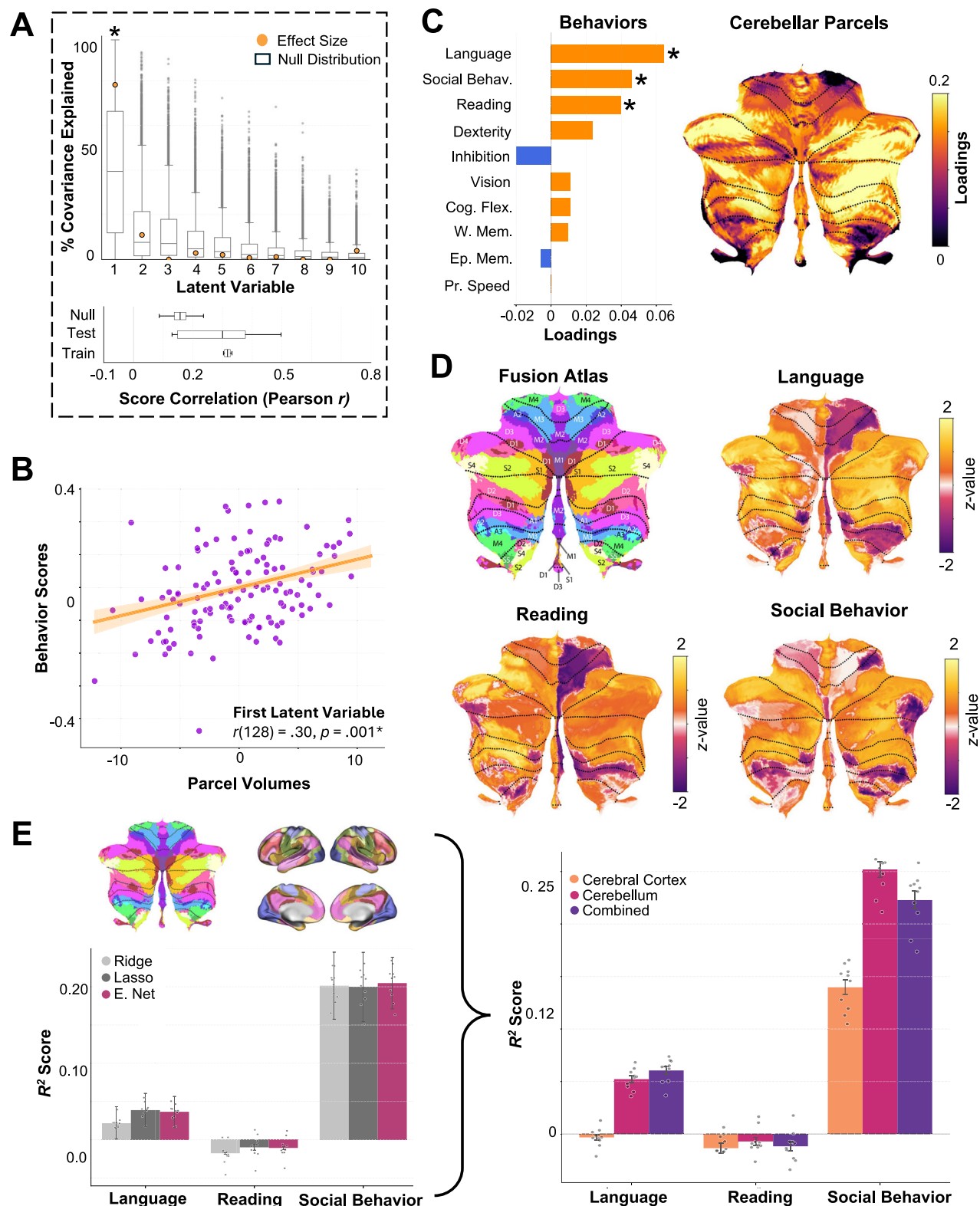

The PLS analysis revealed that language comprehension, social behavior, and reading abilities had the highest loadings among behaviors, while multi-demand (e.g., D2R, D2L, associated with retrieval) and socio-linguistic (e.g., S2R, S4R, associated with social processing and self-projection, respectively) cerebellar parcels showed the highest loadings among parcels. This pattern indicates that the latent variable primarily captured covariation between socio-linguistic behaviors and associative cerebellar regions. Notably, cerebellar

regions associated with action and sensorimotor functions (e.g., A3, M4, associated with motor imagery and lower limb movements, respectively) exhibited low loadings, suggesting limited covariation with socio-linguistic behaviors. Language comprehension, social behavior, and reading abilities demonstrated significant loadings with 95% CIs not crossing zero after 10,000 bootstrap iterations (Fig. 4C; Supplementary Fig. 15). All cerebellar parcels showed significant loadings during bootstrapping (Supplementary Fig. 15). However,

**Fig. 4 | Association of cerebellar parcel-wise cerebellar growth (fusion atlas) and behavioral performance. A** Top: Partial Least Squares (PLS) analysis identified a significant latent variable capturing 78% of the covariation between parcel-wise normative z-scores and behavioral scores, with significance assessed using permutation testing ($p = 0.010$, 10,000 permutations, one-sided). Box plots show the median (center line), interquartile range (box), and whiskers extending to the most extreme values within 1.5 × IQR of the permutation-derived null distribution (10,000 permutations). Grey points denote individual permutation samples. Bottom: 10-fold cross-validation was applied to the PLS model, and the statistical significance of the held-out sample association (Pearson correlation) was evaluated using permutation testing against a null distribution assuming no cerebellum–behavior relationship ($p = 0.001$, 10,000 permutations, one-sided). Box plots show the median (center line), interquartile range (box), and whiskers extending to the most extreme values within 1.5 × IQR. **B** Empirical correlation for the first PLS component demonstrated a significant positive relationship between cerebellar parcel scores and behavior scores ($p = 0.001$, 10,000 permutations, one-sided). Data points signify individual subject scores. The solid line represents the least-squares regression fit, and the ribbon the 95% CI. **C** Cerebellar parcel-wise and behavioral loadings onto the latent variable. Asterisks signify significance of behavioral performance (CIs not crossing zero) during 10,000 iterations of bootstrap resampling. All parcel loadings were significant during bootstrapping. Parcel loadings are projected on the cerebellar flatmap. **D** Parcel-wise z-scores and standardized age and sex were used to predict behavioral performance in mass-univariate linear regression. This analysis was performed to map developmental behavioral associations directly onto individual cerebellar parcels. Only significant associations (z-scored regression betas) are plotted ($p < 0.05$, two-sided). **E** Cerebellar (fusion), cerebral (Yeo et al. 17-networks), and combined cerebello-cerebral parcel z-scores were compared for how well they predicted socio-linguistic behaviors using ElasticNet regularization. Left: Mean Lasso, Ridge, and ElasticNet model performance ($R^2$) across cerebellar, cerebral, and combined models for each behavior, averaged over 10-fold cross-validation. Error bars represent the standard error of the mean. Right: ElasticNet model performance ($R^2$) for cerebellum-only, cerebral-only, and combined model across language, reading abilities and social behavior, averaged over repeated 10-fold cross-validation. Data points denote repeat-level cross-validation performance. Error bars represent the standard error of the mean. The functional fusion atlas was adapted from Nettekoven et al.[6]. Source data are provided as a Source Data file. E. Net ElasticNet, Social Behav. Social Behavior, Cog. Flex. Cognitive Flexibility, W. Mem. Working Memory, Ep. Mem. Episodic Memory, Pr. Speed Processing Speed, L left, R right.

**Table 2 | Wilcoxon signed-rank comparison of ElasticNet performance for cerebellum vs. other models**

| Behavior | Comparison | $M_{Cerebellum}$ | $SD_{Cerebellum}$ | $M_{Other}$ | $SD_{Other}$ | W | N | p | r |
|---|---|---|---|---|---|---|---|---|---|
| Reading | Cerebellum vs. Cerebral Cortex | −0.01 | 0.01 | -0.02 | 0.01 | 7.0 | 10 | 0.030* | 0.65 |
| | Cerebellum vs. Combined | −0.01 | 0.01 | -0.01 | 0.01 | 11.0 | 10 | 0.105 | 0.52 |
| Language Comprehension | Cerebellum vs. Cerebral Cortex | 0.05 | 0.01 | -0.01 | 0.02 | 0.0 | 10 | 0.002* | 0.87 |
| | Cerebellum vs. Combined | 0.05 | 0.01 | 0.06 | 0.01 | 72.0 | 10 | 0.210 | 0.87 |
| Social Behavior (SRS) | Cerebellum vs. Cerebral Cortex | 0.25 | 0.02 | 0.14 | 0.02 | 0.0 | 10 | 0.002* | 0.87 |
| | Cerebellum vs. Combined | 0.25 | 0.02 | 0.22 | 0.03 | 0.0 | 10 | 0.002* | 0.87 |

* = $p < 0.05$, two-sided.

multi-demand and socio-linguistic parcels exhibited the highest bootstrap ratios (BSR = variable mean loading / standard error across iterations), indicating both strong covariation with socio-linguistic behaviors and high reliability[61] (Supplementary Table 9). PLS results for socio-linguistic behaviors were further supported by mass-univariate linear regression that used parcel z-scores and standardized age and sex to predict behavioral scores. This analysis implicated primarily associative cerebellum regions (social and multi-demand regions in the fusion atlas) in individual differences in socio-linguistic behaviors during development (Fig. 4D). However, it is important to note that this analysis was performed post-hoc to decompose the PLS findings at the individual parcel level, since PLS does not identify specific univariate parcel-behavior associations[62].

Finally, we evaluated the relative contributions of cerebellar versus cerebral (Yeo et al. 17-network atlas[50]) normative development to socio-linguistic behavioral performance using ElasticNet regularized regression (see "Methods": "Association with behavioral performance"). We compared how well normative z-scores from cerebellum-only, cerebral-only, and combined cerebello-cerebral models predicted individual variations in language comprehension, social behavior, and reading abilities, using ElasticNet predictive performance ($R^2$) averaged across a repeated 10-fold cross-validation procedure. Across all three behavioral domains, cerebellum-only models consistently outperformed cerebral-only models (pairwise Wilcoxon signed-rank tests across 10 cross-validation repeats, all two-sided $p < 0.05$). For reading abilities and language comprehension, cerebellum-only models performed comparably to combined cerebello-cerebral models (all two-sided $p > 0.05$), indicating no additional benefit of incorporating cerebral features. In contrast, for social behavior as measured by the Social Responsiveness Scale[31] (SRS; higher scores indicate greater autistic traits), cerebellum-only models significantly outperformed both cerebral-only and combined

models (cerebellum vs. cerebrum: $W = 0.0$, $N = 10$, $p = 0.002$, $r = 0.87$; cerebellum vs. combined: $W = 0.0$, $N = 10$, $p = 0.002$, $r = 0.87$; all two-sided) (Fig. 4E and Table 2). Importantly, the cerebellar predictive performance was the highest for social behavior compared to other behaviors. This suggests a pronounced association between individual differences in normative cerebellar development and (atypical) social behavior across the developmental span from childhood to adulthood, over and above development of the cerebral cortex. These patterns remained consistent when using an anatomical functional parcellation of the cerebral cortex (DK atlas[52]), further highlighting the robustness of our findings (Supplementary Fig. 16 and Supplementary Table 10).

Together, these findings indicate that normative variation in cerebellar development, particularly within posterior association regions, is related to individual differences in socio-linguistic behaviors. Interestingly, cerebellar parcels outperformed cerebral-only models across socio-linguistic behaviors, with the strongest effects observed for (atypical) social behavior, where they also outperformed combined cerebello-cerebral models. This converging evidence across multivariate and univariate approaches supports a selective relationship between cerebellar maturation and individual variation in socio-linguistic abilities, especially social behavior.

## Discussion

The present study describes a comprehensive normative framework for cerebellar development from the first years of life through adulthood. Across lobular, task-based, and resting-state parcellations, cerebellar association regions demonstrated steeper growth trajectories compared to sensorimotor regions. Cerebellar maturation aligned with functionally corresponding cerebral regions, with cerebellar association regions linked to cerebral association regions, and sensorimotor regions linked to cerebral sensorimotor regions.

Furthermore, developmental changes in posterior cerebellar regions associated with socio-linguistic and multiple-demand functions predicted individual differences in socio-linguistic abilities. This relationship was especially strong for social behavior, where cerebellar maturation provided predictive value beyond that of the cerebral cortex, demonstrating the cerebellum's unique contribution to individual differences in social abilities.

The consistent sensorimotor-association differences in growth patterns across both lobular and functional parcellations mirror known patterns of cerebellar evolutionary expansion[63,64] and are in line with previous cerebellar developmental research[28–30,65]. Specifically, our findings align with recent normative modeling studies of cerebellar development, which likewise reported steeper volumetric growth trajectories in association regions relative to sensorimotor territories[28,29]. However, the present study extends this literature in several important ways. First, we establish a comprehensive reference of cerebellar lobular growth spanning the full developmental period from age one through early adulthood. Second, we systematically validate these developmental patterns across multiple cerebellar parcellation frameworks, allowing us to distinguish robust organizational principles from parcellation-dependent effects. Third, we explicitly link cerebellar growth trajectories to concurrent structural maturation of the cerebral cortex, providing insight into coordinated cerebello–cerebral development. Finally, we show that normative cerebellar growth trajectories, particularly in posterior association regions, exhibit domain-specific associations with socio-linguistic behavior that are comparable to or stronger than those observed in the cerebral cortex. Together, these findings advance the growing literature on cerebellar development by positioning cerebellar maturation as a central organizing axis for behavioral development and the formation of distributed cerebello-cerebral networks across childhood and adolescence.

All cerebellar parcels, lobular and functional, exhibited age-related volumetric increases, as expected for cerebellar growth during childhood. Male participants demonstrated greater age-related effects across parcellations. This is in line with reports of sex differences in cerebellar development[28,30,66,67] and parallels patterns observed in cerebral cortical development[24,25]. The observed steep developmental growth trajectories in association regions across parcellations could reflect age-related improvements in underlying cognitive functions commonly associated with these regions, suggesting a contribution of the cerebellum to the development of these processes[4,6,68]. Interestingly, the lobular parcellation revealed systematic growth across both posterior and anterior cerebellar regions, particularly in anterior lobule IV. This widespread pattern reflects the extensive structural changes characterizing early postnatal cerebellar maturation[69], extending beyond regions alone. The developmental specificity of these changes is evident in lobular normative models in the HCP-D dataset (excluding 0–5-year-olds from BCP), where maturation was restricted to posterior regions (Supplementary Fig. 13A). This contrast suggests that rapid anterior-posterior developmental changes occur predominantly during the first postnatal years, consistent with longitudinal findings demonstrating accelerated growth in both anterior and posterior lobules during this period[30]. These developmental trajectories establish critical baselines for typical cerebellar development and might contribute to the detection of neurodevelopmental disorders at various developmental stages.

The three cerebellar functional atlases converged on a consistent developmental organization, characterized by flatter trajectories in sensorimotor regions and steeper trajectories in association regions. At finer spatial scales, however, modest discrepancies emerged. The MDTB and functional fusion atlases showed highly consistent developmental profiles, including stronger age-related effects in left-lateralized cognitive regions associated with linguistic, attentional, and working memory processes. In contrast, the

7-network resting-state atlas exhibited more bilaterally distributed age effects in dorsal attention and default mode networks, likely reflecting its relatively coarse parcellation and bilateral parcel distribution, thereby limiting sensitivity to lateralized effects. Parcellation-dependent differences were also evident in sex-related effects, with sex differences being most pronounced in the fusion atlas, likely reflecting its finer-grained, multimodally-defined functional boundaries. Together, these findings indicate that finer-grained inferences regarding lateralization and sex differences might depend on atlas resolution and functional specificity. However, the consistency in sensorimotor and cognitive developmental patterns across different parcellation frameworks suggests that these patterns represent a robust and biologically grounded axis of cerebellar developmental organization[70].

The pronounced influence of age on cerebellar association regions may suggest that the cognitive functions these areas support undergo substantial refinement during childhood. This is in line with prior studies showing a relationship between brain development and the early-life emergence of cognitive functions such as language[71,72], executive control[73,74], and social cognition[75,76], including within the cerebellum[32]. Because these regions show greater age-related changes relative to anterior sensorimotor regions, they may also be more susceptible to developmental perturbations linked to neurodevelopmental or psychiatric conditions[4,15,21,69]. This is in line with findings showing that cerebellar lesions sustained in infancy are associated with more severe and lasting neurocognitive impairments than those occurring later in life[18], emphasizing the critical role of intact cerebellar development in supporting optimal behavioral and cognitive outcomes. Our normative models, especially when integrated with clinical data, may help identify sensitive periods for the development of different cognitive functions, thus informing early intervention strategies in at-risk pediatric populations[25].

Cerebellar parcels exhibited maturational covariance with cerebral regions implicated in the same functions. For instance, cerebellar regions associated with default-mode and social processing displayed strong associations with corresponding default-mode cerebral cortical areas. Conversely, cerebellar regions subserving sensory functions (e.g., spatial processing) showed coordinated development with sensory cerebral areas (e.g., visual and somatomotor networks). These findings extend previous cerebello-cerebral covariance patterns observed in adults[35] by demonstrating domain-specific co-maturation in development. This aligns with established theories of *developmental diaschisis*, wherein the cerebellum critically influences cerebral maturation[21] through extensive reciprocal connectivity with the cerebral cortex[8–11]. Disruption of cerebellar function during critical developmental windows may consequently impair the maturation and function of connected cerebral regions[69,77,78]. Within this framework, our findings lay the foundations for future research to identify specific cerebral regions vulnerable to dysfunction when particular cerebellar regions are disrupted during development, based on their coordinated maturation patterns.

Furthermore, developmental variation in cerebellar circuits appears linked to individual differences in cognitive and social behaviors. We found that normative growth trajectories in posterior cerebellar regions, particularly functional subdivisions of the posterior-lateral Crus I-II, were associated with behavioral performance in language comprehension, reading, and social behavior. These findings align with growing evidence that the cerebellum, and Crus I-II in particular, plays a central role in non-motor functions such as language[79,80] and social cognition[7,81,82], including during development[18,32,83]. Additionally, cerebellar parcels uniquely contributed to socio-linguistic behavioral variance. Growth patterns in posterior cerebellar association regions exhibited strong associations with individual differences in socio-linguistic behaviors, and particularly social functioning as measured by the SRS[31], which exceeded associations with the cerebral

cortex. Importantly, in the present typically developing sample, SRS scores reflect subclinical variation in social functioning rather than diagnostic status. We here consider SRS scores as indexing individual differences in social abilities across the typical development spectrum and interpret them as a continuous behavioral measure rather than a clinical indicator. While subclinical score ranges may attenuate observed effects, the observation of significant cerebellar–behavior associations within this constrained distribution suggests that cerebellar developmental trajectories are particularly relevant for individual differences in social functioning even within typical development. This pattern resonates with the well-documented involvement of the cerebellum in the neurobiology of autism[17,21,77,84,85] and converges with recent normative modeling work linking structural anomalies in Crus I-II to autistic traits[28,29]. Together, these results suggest that structural maturation of the regions in the posterior cerebellum is not only integral to the typical development of complex socio-cognitive functions but may also serve as a sensitive biomarker of autism-related phenotypes, potentially even more so than the cerebral cortex. These findings reinforce the cerebellum's emerging role as a critical contributor to cognitive development, over and above the cerebral cortex, and highlight its relevance for early identification and monitoring of neurodivergent social trajectories.

Our study also has important limitations. First, our sample size is on the smaller end of population-wide normative modeling endeavors. This is partly attributable to the fact that, to date, there is no reliable cerebellar segmentation method that covers the entire developmental range from the perinatal period to adulthood. Different algorithms (e.g., SUIT[86,87], ACAPULCO[43]) perform variably across age ranges, requiring laborious manual correction. Second, minimal age overlap between HCP-D and BCP datasets means age effects may be confounded with site effects, potentially limiting generalizability. Despite these constraints, our findings use careful quality control and manual correction of cerebellar segmentations to provide a comprehensive framework for cerebellar development from late infancy through adulthood, bridging previous studies that examined infant[30] and child/adolescent populations[28,29] separately. Our publicly available normative models can be further validated and expanded as new data becomes available[25,28]. Future efforts should develop reliable cerebellar functional parcellation methods spanning the entire developmental trajectory, which will enable more robust normative modeling and accurate assessment of individual deviations across all developmental stages.

A further limitation is our application of adult-based functional cerebellar parcellations to developing populations. Even though children and adults may recruit similar cerebellar regions for cognitive functions as early as around the third year of life (e.g., Theory of Mind[32]), functional boundaries in the cerebellum likely vary as a function of age[32]. Here, we restricted our functional atlas modeling to children and adolescents between 5 and 21 years. From this age onward, functional similarity to adult cerebellar organization is considered sufficient to support the use of adult-based parcellations[47–49]. However, this approach assumes that functional organization in the cerebellum remains stable across development, which likely does not fully capture age-related changes in cerebellar functional topography[49]. Moreover, behavioral data in the 5–7 year age range were limited, which may have reduced sensitivity to detect age-specific effects. Future research should develop age-specific functional atlases of the cerebellum across different tasks and developmental stages, ideally with larger samples, to enable developmentally appropriate parcellation schemes that could reveal age-dependent spatial changes in cerebellar functional organization.

Lastly, in this study, we focused on establishing normative cerebellar growth trajectories in a typically developing cross-sectional cohort. As such, the reported developmental trajectories reflect age-related variation at the cohort level and cannot determine how a given individual brain changes across development. While this design does not allow the inference of timing-sensitive, causal, or mechanistic processes underlying cerebellar–cerebral maturation, these normative models nevertheless provide a necessary reference framework for future work aimed at elucidating such mechanisms. Longitudinal studies will be required to determine when deviations from normative trajectories first emerge, how cerebellar and cerebral maturation dynamically influence one another within individuals, and whether specific periods of cerebellar vulnerability or plasticity can be identified. Future research should therefore validate and extend our models in longitudinal or clinical populations to assess individual variability over time[88] or in or in neurodevelopmental and psychiatric conditions. For example, comparing cerebellar subregional volumes for autistic individuals to our reference model could reveal its utility in the early identification of autism. In the same vein, it can be assessed whether cerebellar or cerebello-cerebral deviations from the reference model predict individuals' and patients' behavioral outcomes. At later stages, prognostic brain scans in young children may be used to predict increased risk for several disorders or behavioral deficits. Such applications could enable early detection of cerebellar abnormalities and inform targeted interventions. The ability to quantify individual phenotypes relative to age-matched normative reference ranges represents a crucial step toward precision medicine approaches in pediatric neurology and psychiatry, particularly for conditions where cerebellar involvement is suspected or established[25].

Overall, this study establishes normative references for human cerebellar development, providing critical reference benchmarks for detecting atypical cerebellar maturation from late infancy through adulthood. Our results reveal that cerebellar posterior association regions systematically demonstrate prolonged growth trajectories that reflect their essential contributions to cognitive and especially social functions. The domain-specific coordination between cerebellar and cerebral maturation, combined with the predictive capacity of cerebellar volumes for social cognitive and linguistic behaviors, highlights the cerebellum's fundamental role in neurodevelopment and its heightened alteration in autism. By characterizing the relationship between cerebello-cerebral structural co-maturation and behavioral development, our work deepens understanding of cerebellar development relative to cerebral and behavioral development. These findings provide a foundation for precision medicine approaches in pediatric neuropsychiatric conditions and neurodiverse profiles involving cerebellar dysfunction.

## Methods
### Participants
We leveraged openly available cross-sectional MRI and behavioral data of typically developing infants, children, and adolescents from the BCP[41] and the HCP-D[42] datasets. BCP contains structural and functional MRI data of 398 healthy term-born infants from birth to five years of age. HCP-D contains structural and functional MRI data of 652 healthy children and adolescents from five to twenty-one years of age. Informed consent was obtained from adult participants or the parents of underage participants before participation, and all data acquisition procedures complied with the ethical regulations of the associated institutions in the original studies. Exclusion criteria include preterm birth, complications at birth, and medical, genetic, neurodevelopmental, or endocrine conditions. A detailed list of inclusion and exclusion criteria is provided in refs. 41,42. In the present study, we conducted our analyses starting at 12 months of age as the cutoff point. In these scans, cerebellar lobular folds were visible and image processing and segmentation algorithms performed reasonably well (Supplementary Fig. 3), verified by visual inspection among three independent reviewers per scan (AM, NM, LA), and supervised by the project's Principal Investigator (SLV). Overall, we excluded 94 BCP and 12 HCP-D participants due to poor image quality, insufficient cerebellar

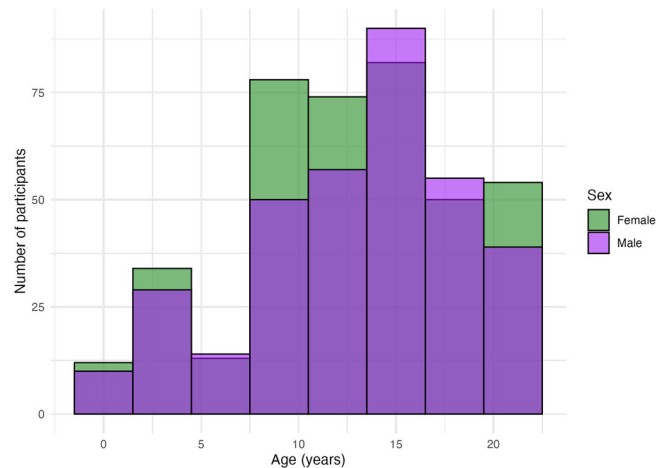

**Fig. 5 | Age distribution of participants by sex in the final sample.** Overlapping bars indicate the number of male and female participants within each age bin. Source data are provided as a Source Data file.

coverage, or poor performance of the segmentation algorithms after visual inspection among the same three independent reviewers. We further excluded 11 participants due to artifacts in cerebral cortical surface reconstruction (see "Cerebral cortex parcellation"). This resulted in a final sample of 751 participants (age: $M = 12.86$, $SD = 5.49$; 397 female): 111 participants from BCP (age: $M = 2.93$, $SD = 1.36$; 53 female) and 640 participants from HCP-D (age: $M = 14.44$, $SD = 4.06$; 344 female). See Fig. 5 for a distribution of the final sample per age and sex.

### Data acquisition
All data were acquired with 3 T Siemens Prisma MRI scanners (Siemens, Erlangen, Germany) with 32 channel head coils. Younger participants in the BCP were scanned while naturally asleep without the use of sedatives. T1-weighted (T1w) images were acquired with a 3D MPRAGE sequence with the following parameters: sagittal field of view of $256 \times 240 \times 166$ mm with a matrix size of $320 \times 300 \times 208$ slices, resolution of 0.8 mm isotropic voxels, and flip angle of 8 degrees. The TR/TE parameters for BCP and HCP-D T1w images are 2400/2.24 ms and 2500/2.22 ms, respectively. Full acquisition parameters and scanning procedures are described in refs. 41,89.

### Behavioral tasks
We examined associations between normative volumetric growth in each parcel of the functional fusion atlas and task performance in the HCP-D dataset, which provides participant scores across multiple behavioral domains. We focused on tasks that were representative of diverse sensorimotor and cognitive domains and optimized for the entire HCP-D age range (5-21 years): i) the Pegboard Test for fine motor dexterity; ii) the Visual Acuity Test for visual processing; iii) the Picture Sequence Memory Task for episodic memory; iv) the List Sorting Working Memory Test for working memory; v) the Dimensional Change Card Sort Task for cognitive flexibility; vi) the Flanker Inhibitory Control and Attention Test for executive function and inhibition; vii) the Pattern Comparison Processing Speed Test for processing speed; viii) the Oral Reading Recognition Test for reading ability; ix) the Picture Vocabulary Test for language comprehension; and x) the Social Responsiveness Scale for social behavior[31]. Most of these assessments (excluding the Social Responsiveness Scale) are components of the National Institutes of Health (NIH) Toolbox (https://nihtoolbox.org/). Detailed descriptions of all tasks are available at: https://humanconnectome.org/storage/app/media/documentation/LS2.0/LS_2.0_Release_Appendix_2.pdf. All scores were age-adjusted.

Distributions of participant scores for each behavioral task can be found in Supplementary Fig. 14.

### Image preprocessing
All processing started with T1w images in participants' native space. HCP-D images were already preprocessed according to the HCP minimal preprocessing pipeline[90]. Specifically, we utilized native space T1w images that had been aligned to T2w images and undergone bias field correction. For BCP images, we adopted a modified approach, given that these data are not currently provided in preprocessed format. BCP preprocessing steps were chosen to be consistent with the HCP-D preprocessing pipeline. For each BCP participant, one of three independent reviewers (AM, NM, LA) conducted visual quality assessment of all scans, evaluating imaging artifacts, motion artifacts, and cerebellar coverage. Each individual-reviewer assessment was subsequently examined by the other two reviewers for consensus validation. 10% of images were then visually inspected by the project's Principal Investigator (SLV) to ensure further validation. Quality-checked BCP images were then minimally preprocessed using iBEAT V2.0[91], a state-of-the-art pipeline specifically optimized for infant and toddler MRI data. T1w and T2w images were reoriented to a consistent left-posterior-inferior (LPI) orientation, underwent bias field correction, and were subsequently aligned to each other. Preprocessed T1w images were then submitted to separate pipelines for cerebellar and cerebral parcellation.

### Cerebellar parcellation
Given that lobular and functional boundaries within the cerebellum do not well coincide[4], we parcellated the cerebellum based on both lobular and functional cerebellar atlases. For the lobular segmentation, we used ACAPULCO, a validated method that uses convolutional neural networks (CNNs) to parcellate the cerebellum into its anatomical lobules[43]. The ACAPULCO algorithm was used to segment cerebella in the entire age range (1–21 years; see Supplementary Fig. 3 for example ACAPULCO segmentations in very young children in the BCP data). After registering the T1w images to the 1 mm isotropic MNI ICBM 2009c template[92] (henceforth referred to as MNI), ACAPULCO used a locating CNN to detect and fit a bounding box around the cerebellum. The cropped cerebellum was then used as input to a second parcellating CNN, which divided the cerebellum into 28 bilateral lobules. Both CNNs were trained based on expert lobular labelling in an adult ($N = 15$) and pediatric ($N = 20$) cohort (see Han et al., 2020 for details). Each MNI-based parcellation was then transformed back to original participant space via nearest neighbor interpolation. Each parcellation output underwent visual inspection and potentially manual correction for cerebellar over- and under-inclusions by three independent reviewers (AM, NM, CY) using ITK-SNAP (https://itksnap.org/pmwiki/pmwiki.php)[93] (see Supplementary Fig. 2 for examples of over- and under-inclusions). To minimize potential bias, each reviewer's corrections were subsequently examined by the other reviewers for consensus validation. 10% of manually corrected images were also visually inspected by the project's Principal Investigator (SLV) to ensure further validation.

For the functional parcellations, we leveraged resting-state[36] and task-based[4] atlases of the cerebellum, as well as a recently developed atlas that fuses large-scale task-based and resting-state data from multiple datasets[6]. The resting-state atlas parcellated the cerebellum into known resting-state networks (limbic, somatomotor, ventral attention, dorsal attention, frontoparietal, and default-mode), based on resting-state functional connectivity with the same networks in the cerebral cortex[36]. The task-based (MDTB[4]) atlas used fMRI task contrast from 26 tasks (47 unique conditions) to parcellate the cerebellum. Here we used the MDTB atlas with ten functional regions: (1: Left-hand (motor) presses, 2: Right-hand (motor) presses, 3: Saccades, 4: Action observation, 5: Divided attention (left hemisphere), 6: Divided

attention (right hemisphere), 7: Narrative, 8: Word comprehension, 9: Verbal fluency, 10: Autobiographical recall). Lastly, the fusion atlas[6] parcellated the cerebellum into subregions corresponding to four large-scale functional domains (motor, action, multi-demand, socio-linguistic), derived by combining multiple datasets under a Hierarchical Bayesian framework. Here, we used the symmetric, mid-granularity version of the atlas with each of 32 bilateral regions belonging to one of the four functional domains.

Given the absence of an established functional atlas for the developing cerebellum, we applied functional atlases exclusively to the HCP-D dataset (ages 5–21 years). We did not extrapolate the adult-derived functional atlases to 1–5-year-olds, as functional spatial patterns in the first years of life may differ from those in adults[47–49]. However, such functional patterns appear to become more comparable to those of adults later in childhood, both in terms of task-specific and resting-state activity[47–49]. We calculated cerebellar volumes by resampling MNI-aligned atlases to each participant's native space. This was achieved using nonlinear warp fields generated during spatial normalization (via FNIRT) within FSL's (https://fsl.fmrib.ox.ac.uk/fsl/docs/#/) *applywarp* tool, with nearest-neighbor interpolation to preserve the discrete nature of atlas labels. We performed additional quality control by excluding participants for whom 50% or more parcels per segmentation were considered outliers (parcel volumes > |2| standard deviations of the mean volume of each parcel). Thus, we ended up excluding 13 participants for the resting-state atlas, 21 for the MDTB atlas, and 3 for the fusion atlas.

### Cerebral cortex parcellation
We obtained participant-specific cerebral cortex parcellations using FreeSurfer's (https://surfer.nmr.mgh.harvard.edu/) *recon-all* pipeline to measure the grey matter volume of parcels in the Yeo et al. 17-network resting-state atlas[50] and the DK gyral atlas[52]. The *recon-all* pipeline includes intensity normalization, skull stripping, tessellation of the gray and white matter cerebral boundary, automated topology correction[94], and surface reconstruction. Surfaces were visually inspected and, when necessary, manually corrected as recommended in the FreeSurfer pipeline (https://surfer.nmr.mgh.harvard.edu/fswiki/FsTutorial/TroubleshootingData). Corrected surfaces were then registered to a spherical surface atlas and underwent labelling based on the Yeo et al. 17-network or DK atlases. Gray matter volumes were calculated from the *aparc+aseg* segmentation, where each voxel is classified as belonging to a specific parcel based on probabilistic atlas information and local intensity values. Parcel volumes are computed as the sum of voxels assigned to each atlas region multiplied by the voxel volume. We used the Euler index (EI) to perform additional quality control of surface reconstruction. The EI was defined as the total number of surface "holes" or topological defects across both hemispheres in the cerebral surface reconstruction prior to FreeSurfer's topological correction[24]. Given that the EI does not have a single threshold that is expected to be generalizable across different datasets[95], in this study, we excluded scans with an EI > |2| median deviations from the dataset-specific median EI[24]. This resulted in the exclusion of 11 participants (see "Participants"), who were discarded from subsequent normative modeling analyses.

### Normative modeling
We generated normative models for lobular and functional cerebellar and cerebral cortical subregions using HBR as implemented in the PCNtoolkit version 0.29[25,44,45] in Python 3.11.8. In this way, we were able to obtain normative ranges of cerebellar volumetric growth and model individual heterogeneity (defined as deviation from the normative volumetric trajectory) with possibly substantial behavioral and clinical implications[25,45,96]. Importantly, the HBR framework allows controlling for possible confounds due to pooling data from different acquisition sites by modelling them as (random) batch effects. HBR leverages shared priors, enabling the estimation of site-specific parameters and hyperparameters. These hyperparameters can be transferred to new sites without the need for the original data[26,28], which also allows models to grow as new data becomes available. We estimated normative volumetric trajectories for each cerebellar and cerebral parcel, modelling age as the main predictor of interest and sex and acquisition site as batch effects (see Supplementary Fig. 1 for the distribution of participant ages across sites). We split our data into a training set (80%) and a test set (20%) while stratifying our sampling to ensure that sex and site were proportional among the training and test sets.

We generated linear and third-order B-spline models with five evenly spaced knot points for every parcel. We evaluated out-of-sample predictive performance of both model types using LOOCV with Pareto-smoothed importance sampling. Performance was summarized as the expected log pointwise predictive density (ELPD-LOO). Higher (less negative) values indicate better predictive accuracy. We accommodated skewed distributions through the sinh-arcsinh likelihood (SHASHb)[44] and modeled random effects in intercept, slope, and variance (sigma) on the batch-effects (sex and site). Lastly, we used four Markov chain Monte Carlo chains with 2,000 samples, with the first 500 considered tuning samples and removed from further analyses. We included the same participants in both cerebellar and cerebral normative models to enable associations of cerebellar and cerebral developmental trajectories at the individual level.

### Association with normative trajectories of the cerebral cortex
We examined associations between individual volumetric growth trajectories (normative model-derived participant *z*-scores) of cerebellar (lobular and functional atlases) and cerebral parcels (Yeo et al. 17-network and DK gyral atlases) to investigate maturational coupling. For the functional parcellation analysis, we focused exclusively on the functional fusion atlas[6] given the consistent normative trajectories observed across all functional parcellations. For the lobular parcellation analysis, we restricted our investigation to the HCP-D dataset to derive accurate cerebral parcellations using FreeSurfer (FreeSurfer is not recommended for younger age ranges included in the BCP dataset[97]). This also maintained consistency with the age range employed in the functional fusion cerebellar-cerebral comparison. As explained previously, we did not extrapolate functional atlases derived in adults to individuals younger than five, to prevent unjustly projecting adult brain functions onto early childhood brains[47–49].

We employed regularized regression to examine associations between the normative model *z*-scores of cerebellar and cerebral parcels, predicting cerebral *z*-scores from cerebellar *z*-scores. Regularization introduces penalty terms to the standard regression loss function, placing constraints on model coefficients to reduce the risk of overfitting. We considered three regularization methods: Ridge regression (L2 penalty, which shrinks coefficients toward zero but retains all features), Lasso regression (L1 penalty, which can shrink coefficients to exactly zero for automatic feature selection), and ElasticNet regression (which combines L1 and L2 penalties).

Model selection and evaluation were performed using nested cross-validation to prevent overfitting during hyperparameter selection[98]. In the outer loop, participants were partitioned into 10 folds (90% training, 10% testing). Within each outer fold, hyperparameters were optimized via an inner 10-fold cross-validation on the training data only, testing seven logarithmically spaced regularization strength parameters ($\alpha$) from 0.001 to 1,000 (and five L1 values from 0.1 to 0.99 for ElasticNet). Model performance was evaluated on the held-out test data from the outer loop. In the Yeo et al. 17-network atlas, Lasso regression demonstrated superior performance with an average cross-validated $R^2 = .36$ (Fig. 3C).

 

For the final analysis, cerebellar parcel volumes were used to predict cerebral parcel volumes using Lasso regression with a global $\alpha = 0.1$. This global $\alpha$ was determined via 10-fold cross-validation in a multi-output Lasso model predicting all cerebral parcels simultaneously and aligns with the distribution of parcel-wise optimal $\alpha$ values, where over 80% of cerebral parcels individually favored $\alpha = 0.1$ (see Supplementary Figs. 11, 13 for model selection and optimal $\alpha$ values in the DK and lobular atlases). Statistical significance of individual cerebellar-cerebral associations was assessed via permutation testing: the target variable was shuffled 10,000 times while maintaining the predictor structure, the model was refitted for each permutation, and a null distribution of regression coefficients was generated for each cerebellar-cerebral pair. The observed coefficients were then compared against this null distribution to derive one-sided $p$-values, which were subsequently corrected for multiple comparisons across all parcel pairs using FDR[51] at $q = 0.05$.

### Association with behavioral performance

We used PLS analysis to associate participants' normative model-derived $z$-scores for each cerebellar parcel with their behavioral scores in the HCP-D dataset (see "Behavioral tasks"). PLS is a multivariate technique that identifies orthogonal *latent variables*, pairs of weighted linear combinations of variables, which maximally covary[60]. Here, each latent variable consists of a set of parcel weights, a corresponding set of behavioral weights, and an associated singular value that quantifies the covariance between cerebellar structure and behavioral performance explained by that component. It is important to note that PLS does not establish causal links between cerebellar growth and behavioral development, does not identify specific univariate parcel-behavior associations, and does not rule out the possibility of other relationships between cerebellar parcels and behaviors[62].

Prior to the analysis, behaviors were standardized and residualized for age and sex to be consistent with cerebellar normative $z$-scores. We evaluated up to ten latent variables, with this maximum number determined by the smaller of the two matrix dimensions (i.e., behaviors). For each component, we computed the percentage of total covariance explained. Statistical significance of each component was assessed using permutation testing (10,000 iterations), in which the behavioral data were randomly shuffled while preserving the structure of the cerebellar matrix. This procedure yielded a null distribution of singular values under the assumption of no brain-behavior association. Since only the first latent variable was statistically significant, other latent variables were not analyzed further.

Statistical significance of the empirical correlation between normative parcel $z$-scores and behavioral scores was determined through permutation testing with 10,000 iterations, where behavioral scores were randomly shuffled to create a null distribution assuming no brain-behavior relationships. To assess stability of the PLS approach, we applied bootstrap resampling (10,000 iterations) and calculated 95% CIs around the loading weights of parcel $z$-scores and behaviors (Supplementary Fig. 15). We computed bootstrap ratios (BSR) as the ratio of each loading to its bootstrap standard error across iterations. BSR values approximate $z$-scores, with $|BSR| > 2.57$ indicating significant contribution at $p < 0.01$ (two-sided)[61]. Importantly, BSR values also reflect the relative salience (contribution strength) of each variable to the brain-behavior relationship, allowing us to rank variables by their importance[61] (Supplementary Table 9). Variables were considered significant if (1) 95% CI did not include zero, and (2) $|BSR| > 2.57$. Lastly, we employed 10-fold cross-validation on the PLS model. Specifically, the PLS model was fitted on the training set (90% of the data), and its performance was evaluated on the held-out test set (10% of the data). This procedure was repeated ten times with different random splits to assess the model's generalizability across different subsets of participants. Correlations between cerebellar parcels and behavioral scores were separately calculated in training and test sets.

Finally, we assessed how cerebellar normative growth was associated with each behavior in relation to cerebral growth. Normative $z$-scores from the functional fusion cerebellar models and the Yeo et al. 17-network and DK cerebral models were used as multivariate predictors of individual behavioral measures. Cerebellar and cerebral parcels were analyzed separately and in combination by concatenating their respective features. For each behavioral outcome, we employed regularized regression to associate individual participants' parcel $z$-scores (cerebellum-only, cerebral-only, or combined cerebello-cerebral) with their behavioral scores, capturing distributed brain-behavior relationships. We compared three regularization methods: Ridge regression (L2 penalty), Lasso regression (L1 penalty), and ElasticNet regression (combined L1 and L2 penalties). Model selection and evaluation were performed using the same nested cross-validation procedure described in the previous section ("Association with normative trajectories of the cerebral cortex"), except here the inner 10-fold cross-validation was used solely for hyperparameter tuning, whereas model performance was quantified using a repeated 10-fold outer cross-validation procedure. By repeating the outer cross-validation 10 times, we obtained 10 sets of 10 train–test partitions. Predictive accuracy ($R^2$) was first averaged across folds within each repeat, yielding one performance estimate per repeat, and then summarized across repeats to provide a stable estimate of how well cerebellar, cerebral, and combined normative features predicted behavioral scores. ElasticNet consistently achieved comparable or superior $R^2$ scores relative to Ridge and Lasso regression across all feature sets and behavioral scores (Fig. 4E, left; Supplementary Fig. 16, left) and was therefore selected over other methods. Performance differences between cerebellar-only, cerebral-only, and combined feature sets were evaluated using pairwise Wilcoxon signed-rank tests on repeat-level performance estimates across the ten cross-validation repeats ($N = 10$).

### Reporting summary

Further information on research design is available in the Nature Portfolio Reporting Summary linked to this article.

## Data availability

The present study used existing developmental data from the Lifespan BCP (https://nda.nih.gov/edit_collection.html?id=2848) and the Lifespan 2.0 HCP-D release (https://nda.nih.gov/general-query.html?q=query=featured-datasets:HCP%20Aging%20and%20Development). Neuroimaging and behavioral data are publicly available for download from the provided links. The cerebellar functional atlases are available on GitHub (https://github.com/DiedrichsenLab/cerebellar_atlases). The cerebellar growth models for lobular and functional parcels are also available on GitHub (https://github.com/kmanoli/NormCerebellum). Source data are provided with this paper.

## Code availability

Image preprocessing leveraged open-source software (iBEAT V2.0[91]: https://github.com/iBEAT-V2/iBEAT-V2.0-Docker; HCP minimal preprocessing pipeline[90]: https://github.com/Washington-University/HCPpipelines). Parcellations for the cerebellum and the cerebral cortex were also derived by openly available tools (ACAPULCO[43]: https://gitlab.com/shuohan/acapulco; FSL: https://fsl.fmrib.ox.ac.uk/fsl/docs/#/; FreeSurfer: https://surfer.nmr.mgh.harvard.edu/). ITK-SNAP[93] for manual correction of segmentations is freely available online (https://itksnap.org/pmwiki/pmwiki.php). Normative modeling code is available via the PCNtoolkit[25,44,45] (https://github.com/amarquand/PCNtoolkit). Scripts for all processing and analysis pipelines used in this study can be found on GitHub (https://github.com/kmanoli/NormCerebellum).

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

## Acknowledgements

AM was funded by the German Academic Scholarship Foundation (Studienstiftung des deutschen Volkes) and the Max Planck Society's Lise Meitner Excellence Program (awarded to SLV). SLV and NM were also funded in part by Helmholtz Association's Initiative and Networking Fund under the Helmholtz International Lab grant agreement InterLabs-0015, and the Canada First Research Excellence Fund (CFREF Competition 2, 2015-2016) awarded to the Healthy Brains, Healthy Lives initiative at McGill University, through the Helmholtz International BigBrain Analytics and Learning Laboratory (HIBALL). CGW was supported by an ERC Starting Grant (REPRESENT 101117806). JD and AFM were supported by the Raynor Cerebellum Project. SLV was furthermore supported by the Max Planck Society's Lise Meitner Excellence Program, the Jacobs Foundation Research Fellowship, and the Hector Foundation Research Development Award.

## Author contributions

A.M., N.M., and S.L.V. originally conceptualized and designed the study. A.M. curated, preprocessed, and segmented the data with support from N.M., F.H., T.T., and L.A. A.M., N.M., and L.A. performed image quality control under the supervision of S.L.V. A.M., N.M., and C.Y. performed manual correction of cerebellar segmentations under the supervision of S.L.V. A.M., and N.S. set up and performed the normative modeling analyses. A.M. performed all additional analyses. A.A.A.B. and A.F.M. provided expertise on normative modeling. J.D. provided expertise on functional parcellation of the cerebellum. S.L.V. provided general project supervision. A.M. created the visualizations and wrote the original draft. A.M., N.M., F.H., N.S., T.T., A.A.A.B., L.A., C.Y., M.K., T.M., T.W., C.P., C.G.W., A.F.M., J.D., and S.L.V. contributed to the interpretation of results. A.M., N.M., F.H., N.S., T.T., A.A.A.B., L.A., C.Y., M.K., T.M., T.W., C.P., C.G.W., A.F.M., J.D., and S.L.V. edited and reviewed the final manuscript.

## Funding

## Competing interests

The authors declare no competing interests.

## Additional information

¹Max Planck Institute for Human Cognitive and Brain Sciences, Leipzig, Germany. ²Institute of Neuroscience and Medicine (INM-7: Brain and Behaviour), Research Center Jülich, Jülich, Germany. ³Faculty of Medicine, Leipzig University, Leipzig, Germany. ⁴Institute of Systems Neuroscience, Medical Faculty and University Hospital Düsseldorf, Heinrich Heine University, Dusseldorf, Germany. ⁵Donders Institute for Brain, Cognition and Behavior, Radboud University Nijmegen, Nijmegen, The Netherlands. ⁶Department for Cognitive Neuroscience, Radboud University Medical Center Nijmegen, Nijmegen, The Netherlands. ⁷Centre for Precision Psychiatry, Division of Mental Health and Addiction, University of Oslo and Oslo University Hospital, Oslo, Norway. ⁸Department of Psychology, Faculty of Social Sciences, University of Oslo, Oslo, Norway. ⁹Department of Psychology, Pedagogy and Law, School of Health Sciences, Kristiania University College, Oslo, Norway. ¹⁰Department of Psychiatry and Psychotherapy, University of Tübingen, Tübingen, Germany. ¹¹German Center for Mental Health (DZPG), Jena, Germany. ¹²Cognitive Neuroscience Lab, Department of Liberal Arts and Sciences, University of Technology Nuremberg, Nuremberg, Germany. ¹³Department of Neuroimaging, Institute of Psychiatry, Psychology, & Neuroscience, King's College London, London, United Kingdom. ¹⁴Western Institute of Neuroscience, Western University, London, ON, Canada. ¹⁵Department of Statistical and Actuarial Sciences, Western University, London, ON, Canada. ¹⁶Department of Computer Science, Western University, London, ON, Canada. ✉e-mail: manoli@cbs.mpg.de; valk@cbs.mpg.de

