## [Transparent Peer Review file · Nature Communications]

Cerebellar growth is associated with domain-specific cerebral maturation and socio-linguistic behavior

Corresponding Author: Ms Aikaterina Manoli

Version 0:

Reviewer comments:

Reviewer #1

(Remarks to the Author)

The paper develops a lifespan normative model of cerebellar development from infancy to young adulthood (ages 1–21 years) using MRI data from 751 individuals in the BCP and HCP-D datasets. It examines volumetric cerebellar growth, maturational coupling between cerebellar and cerebral cortical regions, and cerebellar contributions to brain–behavior associations across development. Based on normative modeling and regression analyses, the study finds that posterior cerebellar regions show steeper developmental growth, coordinated maturation with functionally related cerebral areas, and stronger associations with socio-linguistic and social behaviors than anterior sensorimotor regions. While these results are conceptually important, some methodological and interpretative limitations raise concerns about the robustness and generalizability of the conclusions.

1. The study largely reiterates the well-established role of the posterior cerebellum in higher cognition rather than providing new mechanistic or developmental insights into how this relationship emerges over time, such as identifying timing-sensitive mechanisms and exploring functional coupling trajectories.

2. A major methodological limitation is the mismatch between parcellation schemes, as the study applies functional parcellations to the cerebellum but relies on the anatomically defined Desikan–Killiany (DK) atlas for the cerebral cortex, which may obscure functionally specific cerebello–cerebral correspondences and thus limit the interpretability of the reported maturational coupling. As a result, the reported cerebello–cerebral maturational coupling may be blurred or biased, since functionally related regions in the cerebellum could be compared against cortical parcels that do not reflect the same network boundaries. A more appropriate approach would be to use a functionally defined cortical parcellation, such as the Yeo 7- or 17-network atlas.

3. In Figure 3, the left and right DK parcels are averaged in the heatmap to summarize cerebello–cerebral associations. However, it is unclear whether this averaging is appropriate, as it may obscure hemispheric lateralization of the associations. If the coupling is strongly lateralized, the averaging could mask meaningful asymmetries. The authors should evaluate this assumption and consider alternative approaches, such as sigmoid-weighted combination, or maximal-value projection, to better preserve lateralized effects.

4. Although the authors included three functional atlases to ensure that the observed developmental patterns were not specific to a single framework, the task-based, resting-state, and multimodal atlases define cerebellar boundaries differently. As a result, shared regions may show divergent growth, age, and sex effects, likely reflecting parcellation differences rather than true biological variation, as evident in Figure 2. The authors only mention the similarity between them briefly but do not sufficiently explain or compare the discrepancies across the results obtained from different atlases, which may limit the interpretability and biological specificity of the reported developmental patterns.

5. Although the authors employed nested 10-fold cross-validation for model selection and evaluation, the final behavioral prediction analyses were based on a single round of 10-fold CV using approximately 128 participants. Given the small sample size and potential variability of fold assignments, relying on a single split may yield unstable or optimistic estimates of predictive performance. A more rigorous approach would involve repeated cross-validation or external validation on an independent dataset to ensure the robustness and generalizability of the reported brain–behavior associations.

6. The paper also suffers from imprecise descriptions, inconsistent figure scaling, incomplete or unclear captions, and missing units in figures and tables, which together reduce the clarity, readability, and professionalism of the manuscript. The overall writing could be more formal, precise, and logically organized.

(Remarks on code availability)

Reviewer #2

(Remarks to the Author)

Summary

The study presents several noteworthy findings, particularly the detailed characterization of cerebellar growth trajectories across childhood and adolescence and their relationship to cognitive development. The use of both lobular and functional parcellations provides convergent evidence supporting the cerebellum's dual motor and cognitive roles, which aligns well with and extends previous work (e.g., Stoodley & Schmahmann, 2009; Guell et al., 2018). The inclusion of two independent cohorts and robust statistical modeling strengthens the generalizability of the results. Comparisons of maturation with the cerebral cortex are very interesting.

The work is significant for developmental neuroscience and related fields, offering novel insights into how cerebellar maturation supports emerging cognitive abilities. The methodology is sound and consistent with current standards in neuroimaging and developmental modeling. The analyses are clearly described, and the level of methodological detail appears sufficient for replication.

The data support the authors' main conclusions, and the interpretation is generally well balanced. While some aspects (e.g., corpus medullare analyses) would benefit from clarification, these do not undermine the overall findings.

1° Anterior vs. posterior trajectories

(Line 196) The comparison between anterior and posterior lobes could be expanded. Within the posterior lobe, lobule VIIIA is known to contribute to motor functions despite its posterior location (Guell et al., 2018). This is reflected in Figures 2A–B, where the functional parcellation shows a mirror of anterior/motor regions near lobule VIIIA. Supplementary Table 1A shows notably lower beta values for VIIIA than for other posterior lobules. This suggests distinct developmental trajectories between motor and cognitive territories, which may be a more meaningful distinction than an anterior–posterior division.

2° Corpus medullare analyses

The inclusion of the corpus medullare warrants further justification. This region comprises total cerebellar white matter, including deep nuclei, and its overall volume may not be functionally interpretable. It is unclear whether any quality control was performed for this structure, which is difficult to segment accurately. Given its absence from the functional atlas used elsewhere and the large sex differences reported in Supplementary Table 1, the results may reflect segmentation variability. Unless reliability can be demonstrated, these analyses should be removed or clearly qualified.

3° Interpretation of SRS results

The Social Responsiveness Scale (SRS) primarily assesses social functioning in autism spectrum conditions. As the HCP-D dataset excludes individuals with neuro developmental diagnoses, the range of SRS scores is likely limited. Reporting the distribution of SRS scores and commenting on their interpretability in this restricted sample would clarify how meaningful these associations are.

4° Relation to Gaiser et al. (2024)

There is some overlap with Gaiser et al. (2024, Nature Communications), which also used normative modeling in pediatric data and examined SRS associations. Expanding the discussion to highlight similarities, differences, and unique contributions of the current work would improve context and novelty.

5° The abstract states that behavioral results are presented for the entire sample (ages 1–21 years), which is inaccurate.

Behavioral data were only available for the HCP-D cohort, and this limitation should be clearly specified to avoid confusion.

(Remarks on code availability)

The code is available and could be usable.

Reviewer #3

(Remarks to the Author)

Manoli et al. used normative modelling to map cerebellar volumetric growth in $n=751$ infants and young adults. Further, the team investigated how cerebellar growth is linked to cerebral cortical development and cognitive and behavioral outcomes. An advance of this analysis is that both structural (lobular) and functional parcellations were used to evaluate the data, which provides a finer-grained analysis that accounts for the relatively large size of some cerebellar lobules and the known medial-to-lateral functional organization of the cerebellum.

The age effects are consistent with known behavioral / brain developmental trajectories, with sensorimotor regions maturing earlier than those associated with higher cognitive functions. This has been previously established in the cerebellum by other developmental studies (e.g. Tiemeier et al 2010). The link to behavior is a nice addition to this study, although it is only possible in a more restricted age range (5 yrs and up) due to available data. In the analysis of the co-development of the cerebellar parcels and the cerebral cortical parcels, though a major concern is that the cerebellum was parcellated functionally and the cerebral cortical parcellation was a structural parcellation. This is particularly important in the comparison between how well cerebellar vs cerebral cortical regions correlated with behavioral scores and should be addressed.

Overall, this study adds to the expanding literature regarding cerebellar development and the relationship between cerebellar development and emerging behaviors. The available growth curves will be updated with new data as it becomes available, which is a nice resource for the field. The additional analyses investigating co-development of the cerebellum and cerebral cortex and linking cerebellar development to behaviors take full advantage of the data available and lead to a more comprehensive picture. That said, there are some concerns about the analyses as they stand, which are outlined below.

Questions / comments:

Why did the authors exclude the BCP data from birth-age 1?

Please provide the final sample size per age bin in the main text Methods, after exclusion, so that it is easier to understand how much data is represented at different ages. Were there similar numbers of males / females in each bin?

Data is cross-sectional and not longitudinal, which should be emphasized in the limitations; these are not brains that are followed over time, so we do not know how an individual might track over development. This is an extremely common issue in human longitudinal data, so it is not a "problem" per se, but should be mentioned in the Discussion.

The ACAPULCO pediatric parcellation was trained on the ABIDE dataset, which includes ages 5 and up – how confident are the authors in using this algorithm for the 1-5 year olds? While the authors manually checked and corrected the images, this may be a challenge to do on a large scale for other studies. Were more corrections needed for the younger age ranges?

The MDTB task battery parcellations are based on data from young adults, who are outside the age range of this study – though the authors cite other studies showing similarities in parcellations across ages (particularly in older children and adolescents), are these functional parcellations appropriate for the younger age range in this study? Would a voxel-based analysis provide a less constrained approach?

"Our results show that posterior cerebellar regions involved in higher association processes undergo more rapid developmental growth than anterior sensorimotor regions" – the authors should emphasize that these findings are in this age group and that they are missing the 0-1 year old data, which is an age range when there is rapid development in sensorimotor networks and behaviors.

Major concern: Why was the DK atlas used for the cerebral cortical parcellation? The authors parcellated the cerebellum functionally but the cerebral cortex structurally. There are cortical parcellations that reflect e.g. functional networks (such as the Yeo / Buckner maps) that might better reflect co-growth between the cerebellum and cerebral cortex. The Yeo / Buckner parcellations would allow a "like to like" comparison between the cerebral cortex and cerebellum. This is particularly important given the authors' interpretation of the co-development of "functionally similar" cerebral cortical and cerebellar regions.

Why not perform voxel-based analyses? Particularly in the younger ages, performing analyses using parcellations established in adults might not harness the full power of the dataset and the parcellations institute constraints on the data. This approach should be better rationalized, especially in applying the parcellations to a different age group. What is the advantage of this over data-driven approaches which then can be interpreted in the context of the functional parcellations?

The analysis showing how cortical vs cerebellar maturation is associated with individual behavioral scores similarly suffers from the fact that the cortical parcellations are structural and the cerebellar parcellations are functional. Task based cortical parcellation would be a more appropriate comparison here.

p. 20 "The pronounced influence of age on higher association posterior cerebellar regions may suggest that the cognitive functions these areas support undergo substantial refinement during childhood". The authors should reference extensive developmental literature on the emergence and maturation of different behaviors over the course of childhood and adolescence to support this statement in a more specific way. Further, they can evaluate this statement with the data used in the current study.

Minor comments / suggestions:

- Another cerebellar developmental study to add for interpretation / discussion in typically-developing children and adolescents is Moore et al. 2017 Dev Cog Neuroscience. This study showed differential relationships between cerebellar grey matter volume and behavioral scores during development.
- Instead of referring to functional parcels by e.g. "D1" in the main text, a brief description would be helpful for readers e.g. "spatial working memory"; this is done in some places but is not consistent.
- Were the behavioral scores from the HCP the age-adjusted scores or the raw scores?

(Remarks on code availability)

Version 1:

Reviewer comments:

Reviewer #1

(Remarks to the Author)

The authors have substantially revised the manuscript and provided a detailed rebuttal. Several of my major methodological

concerns have been well addressed, particularly regarding the choice of cortical parcellation, the improved robustness of behavioral predictions, and the discussion of atlas-dependent discrepancies.

There are still several concerns that should be addressed to ensure internal consistency and statistical accuracy.

1. About Hemispheric averaging in DK heatmap, I appreciate that the authors have now provided hemisphere-specific results in Supplementary Figure 11 and clarified that the averaging in Figure 3D was performed solely for visualization purposes. However, the decision to present averaged data in the main figure creates a logical tension with the revised Discussion. The authors now argue that discrepancies between atlases (e.g., Fusion vs. Resting-State) are partly driven by their differing sensitivity to lateralization, specifically, that the Fusion atlas captures left-lateralized cognitive effects while the Resting-State atlas is more bilateral. If lateralization is a key factor explaining the results (or the differences between atlases), averaging the hemispheres in the primary figure risks obscuring the very biological properties the manuscript now highlights. I strongly recommend moving the hemisphere-specific heatmap (currently Supplementary Figure 11) to the main text to replace the averaged plot. Alternatively, if the averaged plot is retained, please quantify the correlation between left and right hemispheric patterns to statistically justify that averaging does not mask significant asymmetries.
2. The rebuttal correctly describes the implementation of a 10x10 repeated outer cross-validation, resulting in 100 total fold partitions. The authors state that they performed Wilcoxon signed-rank tests using $N=100$. However, in a repeated cross-validation framework, the 100 folds are not statistically independent samples; the same data points are reused across the 10 repeats. Treating 100 as the sample size for the significance test inflates the degrees of freedom and may yield overly optimistic p-values. Please clarify the sampling unit used for inference. A more conservative and standard approach would be to aggregate performance metrics within each of the 10 full cross-validation repeats (resulting in $N=10$) or to use a corrected t-test designed for repeated cross-validation. Please ensure the reported N , degrees of freedom, and p-values in the text and Table 2 reflect the dependency structure of the repeated folds.
3. There appears to be a discrepancy in the reported regularization parameters. The revised figure caption for Supplementary Figure 12 states that "A global $\alpha = 0.1$ was selected" for the Lasso model. However, the caption text provided in the Supplementary Figure 14 describes a "global $\alpha = 1000$ " for the same analysis. Please review the manuscript and supplementary captions to resolve this discrepancy and ensure the correct parameter value or mathematical symbol is reported.
4. I note that data from birth to 1 year of age were excluded due to segmentation limitations. While this exclusion is methodologically sound given the imaging constraints, the manuscript continues to use the broad term "infancy" in the Abstract and overarching descriptions. Since "infancy" typically encompasses the first year of life, using the term without qualification may be imprecise. Please consider refining the terminology (e.g., "from late infancy" or "from 1 year of age") to accurately reflect the specific developmental window covered by the normative models.

(Remarks on code availability)

While the repository provides a well-structured analysis workflow for cerebellar normative modeling, it would benefit from a version-pinned environment file (e.g., `environment.yml` or `requirements.txt`) to improve reproducibility and ease of use.

Reviewer #2

(Remarks to the Author)

The authors have addressed all my concerns. In my opinion, this paper is suitable for publication.

(Remarks on code availability)

The code is usable

Reviewer #3

(Remarks to the Author)

The revised manuscript has improved in several key areas noted by the reviewers, including a more critical and comprehensive assessment of the limitations of the study and better couching of the study in the context of the prior literature (e.g. recent work by Gaiser et al.).

That said, two of the three reviewers noted that comparing structural cortical parcellations with functional cerebellar parcellations is problematic. The authors conducted the recommended cortical functional parcellation analyses, but these have been pushed into supplement – it seems that that should be the primary analysis for cerebello-cortical coupling, or at the very least for assessing the relative contributions of the cerebral cortex and the cerebellum to behavioral scores. In the abstract, it is stated that "cerebellar and cerebral areas with similar functional roles demonstrate coordinated maturation", but this is better tested with similar parcellations in both structures. Further, the comparison between cerebellar predictors of socio-linguistic behaviors and cortical predictors is using functional cerebellar parcels and structural cortical parcels, with the functional cortical parcels in the supplement; in this case, the cerebellum still shows additional predictive power over the cerebral cortex, but it is less marked than when the cortex is parcellated anatomically. These are important considerations.

Other things to be addressed:

- In the abstract, the phrasing "growth ... aligns with cerebral development and behavioral outcomes" implies a longitudinal analysis, where change over time reflects later behavior. More appropriate phrasing would be something like "how it corresponds with cerebral cortical development and brain-behavior relationships during childhood through adulthood". Similarly, it is recommended that "outcomes" in the title be removed, given that this is not a longitudinal cohort – the

cerebellar growth index is associated with socio-linguistic behavior at the same timepoint, rather than behavioral outcomes. This may sound fussy, but it these edits are important to accurately reflect the analyses.

- Given that the authors have excluded data from birth-age 1, the data do not include the infancy period, which ends at 1 year of age. This should be clarified throughout the manuscript. There also is a limited amount of data before age 7, which should be noted in the limitations, and the functional and cerebro-cerebellar analyses are only conducted in children aged 5+. This is a limitation of the dataset and not by design, but again the edits are needed to accurately report the study.

(Remarks on code availability)

Response to Reviewers - NCOMMS-25-76551-T

We would like to thank the Editor and the Reviewers for their thoughtful and constructive evaluations and for the opportunity to submit a revised version of our manuscript. We believe that the comments and suggestions have substantially improved the clarity, rigor, and interpretability of the work. In response, we have implemented three major revisions.

First, we repeated the analyses of cerebello–cerebral associations using the Yeo et al. 17-network resting-state cerebral cortical atlas to provide a functional alternative to the gyral Desikan–Killiany parcellation. These analyses yielded highly consistent results, also revealing domain-specific associations between higher-order cognitive and sensorimotor regions of the cerebellum and corresponding cerebral networks. Second, we repeated the analysis comparing the relative contributions of cerebellar and cerebral maturation to behavioral outcomes using the Yeo et al. 17-network atlas. Third, we strengthened the robustness of the brain–behavior analyses by implementing repeated cross-validation. Across all analyses, the results remained stable and consistently demonstrated that cerebellar growth trajectories outperform cerebral-only and combined cerebello–cerebral models in predicting individual differences in social behavior across childhood and adolescence.

We have addressed other changes suggested by the Reviewers by providing additional supplementary figures and clarifications in the manuscript’s Introduction, Methods, Results, and Discussion. Please find our detailed responses below according to the Reviewers’ comments one by one. For the revised manuscript, changes are highlighted in yellow and are also listed in the response.

Reviewer #1 (Remarks to the Author):

The paper develops a lifespan normative model of cerebellar development from infancy to young adulthood (ages 1–21 years) using MRI data from 751 individuals in the BCP and HCP-D datasets. It examines volumetric cerebellar growth, maturational coupling between cerebellar and cerebral cortical regions, and cerebellar contributions to brain–behavior associations across development. Based on normative modeling and regression analyses, the study finds that posterior cerebellar regions show steeper developmental growth, coordinated maturation with functionally related cerebral areas, and stronger associations with socio-linguistic and social behaviors than anterior sensorimotor regions. While these results are conceptually important, some methodological and interpretative limitations raise concerns about the robustness and generalizability of the conclusions.

We thank the Reviewer for their positive evaluation of our manuscript and constructive criticism that significantly enhanced the quality of our work.

1. The study largely reiterates the well-established role of the posterior cerebellum in higher cognition rather than providing new mechanistic or developmental insights into how this relationship emerges over time, such as identifying timing-sensitive mechanisms and exploring functional coupling trajectories.

We thank the Reviewer for this thoughtful comment. We agree that our findings are consistent with the established posterior–anterior functional organization of the cerebellum and the role of the posterior cerebellum in higher cognition. The present study extends this literature by moving beyond regional localization to characterize region-wise developmental coupling between cerebellar and cerebral growth trajectories. Additionally, our analyses link interindividual variation in posterior cerebellar maturation profiles to socio-linguistic behavioral outcomes across development. These findings underscore the importance of the cerebellum for socio-linguistic functioning during development and demonstrate that cerebellar maturation emerges as a predictor of atypical social behavior beyond the contribution of cerebral measures alone. In this way, our work provides a developmental framework for relating normative cerebello–cerebral growth trajectories to cognition, helping to bridge descriptive anatomy and behaviorally relevant development. We have further clarified this in the Introduction:

To investigate cerebellar development and how it relates to that of the cerebral cortex and behavioral outcomes, the present study describes normative models of cerebellar volumetric growth from infancy to young adulthood (p. 5, lines 151-153).

Together, these findings provide novel understanding of how cerebellar development aligns with cerebral maturation and supports emerging higher cognitive abilities, with potential relevance for identifying biomarkers of atypical neurodevelopment (p. 5-6, lines 173-176).

We also agree that our design limits inference regarding timing-sensitive or mechanistic processes underlying these associations, a limitation common to cross-sectional developmental studies. We now explicitly acknowledge this limitation in the Discussion of the revised manuscript and clarify that longitudinal data will be required to delineate these causal mechanisms and developmental trajectories:

Lastly, in this study we focused on establishing normative cerebellar growth trajectories in a typically developing cross-sectional cohort. As such, the reported developmental trajectories reflect age-related variation at the cohort level and cannot determine how a given individual brain changes across development. While this design does not allow the inference of timing-sensitive, causal, or mechanistic processes underlying cerebellar–cerebral maturation, these normative models nevertheless provide a necessary reference framework for future work aimed at elucidating such mechanisms. Longitudinal studies will be required to determine when deviations from normative trajectories first emerge, how cerebellar and cerebral maturation dynamically influence one another within individuals, and whether specific periods of cerebellar

vulnerability or plasticity can be identified. Future research should therefore validate and extend our models in longitudinal or clinical populations to assess individual variability over time⁸⁹ or in or in neurodevelopmental and psychiatric conditions. For example, comparing cerebellar subregional volumes for ASD patients to our reference model could reveal its utility in ASD diagnosis. In the same vein, it can be assessed whether cerebellar or cerebello-cerebral deviations from the reference model predict patients' behavioral outcomes. At later stages, prognostic brain scans in young children may be used to predict increased risk for several disorders or behavioral deficits. Such applications could enable early detection of cerebellar abnormalities and inform targeted interventions. The ability to quantify individual phenotypes relative to age-matched normative reference ranges represents a crucial step toward precision medicine approaches in pediatric neurology and psychiatry, particularly for conditions where cerebellar involvement is suspected or established²⁵ (p. 23-24, lines 590-610).

2. A major methodological limitation is the mismatch between parcellation schemes, as the study applies functional parcellations to the cerebellum but relies on the anatomically defined Desikan–Killiany (DK) atlas for the cerebral cortex, which may obscure functionally specific cerebello–cerebral correspondences and thus limit the interpretability of the reported maturational coupling. As a result, the reported cerebello–cerebral maturational coupling may be blurred or biased, since functionally related regions in the cerebellum could be compared against cortical parcels that do not reflect the same network boundaries. A more appropriate approach would be to use a functionally defined cortical parcellation, such as the Yeo 7- or 17-network atlas.

We thank the Reviewer for this important methodological suggestion and agree that aligning cerebellar and cerebral parcellations using a functionally defined framework is critical for interpretable cerebello–cerebral comparisons. In response, we revised our analyses to use the Yeo et al. 17-network cerebral parcellation, which offers a level of granularity comparable to the cerebellar functional fusion atlas (Nettekoven et al., 2024). Specifically, we fit normative developmental models for each resting-state cerebral network and examined their maturational coupling with cerebellar functional regions. We then compared the relative contributions of cerebellar, cerebral, and combined cerebello–cerebral models to behavioral outcomes using the same analytic framework as in the original manuscript.

Importantly, the results were highly consistent with those obtained using the Desikan–Killiany atlas. At the level of cerebello–cerebral coupling, cerebellar social regions (S2 and S3) showed positive associations with default mode network components in the Yeo et al. 17-network atlas (e.g., network 15). Motor cerebellar regions displayed expected network specificity, with M2 selectively associated with sensorimotor and visual networks (e.g., networks 4 and 1), while M3 was anticorrelated with default mode networks (e.g., network 16). The multi-demand cerebellar region D2, implicated in retrieval processes, showed strong coupling with higher-order association networks, including salience and limbic networks (e.g., networks 7 and 9). These

patterns support the robustness of our findings across anatomical and functional cerebral parcellations.

Supplementary Figure 12. Associations of cerebellar (fusion atlas) and cerebral (Yeo et al. 17-network atlas) growth trajectories. **A.** Functional fusion atlas (adapted with permission from Nettekoven et al.⁶). **B.** Mean effect of age on the Yeo et al. 17-network atlas (top) and example normative trajectory for right network 17 (default Age mode; bottom). Bold lines represent the mean trajectories for each sex. Shaded areas represent the 95.5% confidence interval. **C.** Comparison of Ridge, Lasso, and ElasticNet regularization models based on average R^2 scores (10-fold cross-validation). The Lasso model marginally outperformed the other two and was selected (mean $R^2= 0.34$). Error bars represent the standard error of the mean. **D.** Top: Significant cerebello-cerebral associations (10,000 permutations, FDR corrected at $q = .05$). A global

$\alpha = 0.1$ was selected for associations based on a multi-output Lasso model predicting all cerebral parcels simultaneously and aligns with the distribution of parcel-wise optimal α values, where over 80% of cerebral parcels individually favored $\alpha = 0.1$. Bottom: FDR-corrected weights for cerebellar parcels with the largest number of significant associations in the fusion atlas, projected on the Yeo et al. 17-network atlas. Abbreviations: Coef. = coefficient; E. Net = ElasticNet; VIS = visual network; SMN = somatomotor network; DAN = dorsal attention network; SAL = salience network; LIM = limbic network; FPN = frontoparietal network; TPN = temporoparietal network; DMN = default mode network; M = male; F = female; L = left; R = right.

At the behavioral level, cerebellar models continued to outperform cerebral-only and combined cerebello-cerebral models in predicting social behavior across repeated 10-fold cross-validation, further reinforcing the cerebellum's unique contribution to socio-behavioral development.

Supplementary Figure 16. Cerebellar (fusion), cerebral (Yeo et al. 17-networks), and combined cerebello-cerebral parcel z -scores were compared for how well they predicted socio-linguistic behaviors using ElasticNet regularization. Left: Mean Lasso, Ridge, and ElasticNet model performance (R^2) across cerebellar, cerebral, and combined models for each behavior, averaged over cross-validation. Error bars represent the standard error of the mean. Right: ElasticNet model performance (R) for cerebellum-only, cerebral-only, and combined model across language, reading abilities and social behavior, averaged over repeated cross-validation. Error bars represent the standard error of the mean. Abbreviations: E. Net = ElasticNet.

We have accordingly revised the Methods and added a detailed presentation of these analyses in Supplementary Figures 12 and 16, and Supplementary Table 10 (focusing on comparisons of behavioral predictions for cerebellar and cerebral parcel sets). We further refer to these findings in the Results:

Importantly, these domain-specific patterns remained largely consistent when associating cerebellar trajectories with a resting-state functional parcellation of the cerebral cortex (Yeo et al. 17-network atlas⁶⁰), suggesting that the observed cerebello–cerebral coupling captures stable, biologically meaningful functional relationships that extend beyond a single cerebral parcellation scheme (Supplementary Figure 12) (p. 11, lines 297-301).

These patterns remained consistent when using a resting-state functional parcellation of the cerebral cortex (et al. 17-network atlas⁶⁰), further highlighting the robustness of our findings (Supplementary Figure 16 and Supplementary Table 10) (p. 15, lines 381-383).

3. In Figure 3, the left and right DK parcels are averaged in the heatmap to summarize cerebello–cerebral associations. However, it is unclear whether this averaging is appropriate, as it may obscure hemispheric lateralization of the associations. If the coupling is strongly lateralized, the averaging could mask meaningful asymmetries. The authors should evaluate this assumption and consider alternative approaches, such as sigmoid-weighted combination, or maximal-value projection, to better preserve lateralized effects.

We thank the Reviewer for raising this point and agree that hemispheric lateralization is an important consideration when interpreting cerebello–cerebral coupling. We would like to clarify that left–right averaging is applied only in the Figure 3D summary heatmap for visualization purposes, to maintain readability and avoid an excessively large and dense panel. All statistical analyses, model fitting, quantitative results, and other plots (i.e., cerebral cortex plots in Figure 3D) are conducted using hemisphere-specific cerebral parcels, without any left–right averaging. To ensure transparency, we explicitly state this in the figure caption and provide the full hemisphere-resolved (left and right) heatmap in Supplementary Figure 11. Inspection of the hemisphere-specific results indicates that the key patterns reported in the manuscript are not driven by strong lateralized effects that would be obscured by averaging in the summary figure. We have revised the figure caption and main text in the corresponding Results section to further emphasize this distinction and guide readers to the hemisphere-specific results in the Supplementary Materials:

Results:

*See **Supplementary Figure 11** for hemisphere-specific associations in the heatmap (p. 11, lines 296-297).*

Figure 3 caption:

See Supplementary Figure 11 for left and right parcel weights (p. 13, line 318).

Note that, unlike in the heatmap, cerebral hemispheres are not averaged in this plot (p.13, line 321-322).

4. Although the authors included three functional atlases to ensure that the observed developmental patterns were not specific to a single framework, the task-based, resting-state, and multimodal atlases define cerebellar boundaries differently. As a result, shared regions may show divergent growth, age, and sex effects, likely reflecting parcellation differences rather than true biological variation, as evident in Figure 2. The authors only mention the similarity between them briefly but do not sufficiently explain or compare the discrepancies across the results obtained from different atlases, which may limit the interpretability and biological specificity of the reported developmental patterns.

We thank the Reviewer for this insightful comment. In the revised manuscript, we have expanded both the Results and Discussion to more explicitly compare similarities and discrepancies across the three functional atlases and to clarify their interpretation. Specifically, we now distinguish between parcellation-dependent differences at finer spatial scales and developmental patterns that converge across atlases.

Across all three parcellation frameworks, we observed a consistent large-scale developmental organization, characterized by flatter age-related trajectories in sensorimotor regions and steeper trajectories in higher-order cognitive regions. At finer spatial scales, however, modest discrepancies emerged. The task-based MDTB atlas and the functional fusion atlas showed highly consistent developmental profiles, including stronger age-related effects in left-lateralized cognitive regions associated with linguistic, attentional, and working memory processes. In contrast, the 7-network resting-state atlas exhibited age effects distributed more bilaterally in dorsal attention and default mode networks. This likely reflects its relatively coarse parcellation and bilateral parcel definitions, which limit sensitivity to hemispheric specialization. Parcellation-dependent differences were also evident for sex-related effects, which were most pronounced in the fusion atlas, consistent with its finer-grained, multimodally defined functional boundaries.

We now explicitly discuss that these discrepancies are most parsimoniously explained by differences in atlas construction, spatial resolution, and functional boundary definitions, rather than by possible inconsistencies in underlying biology. Importantly, prior frameworks emphasize convergence across different parcellations as a criterion for biological plausibility (e.g., Eickhoff et al., 2018). Hence, we interpret the consistency of sensorimotor versus cognitive developmental trajectories across all atlases as reflecting a robust and biologically grounded axis of cerebellar developmental organization.

Results:

Despite overall consistency in flatter growth trajectories for sensorimotor regions and steeper trajectories for cognitive regions, we observed modest differences across corresponding areas between parcellation schemes. In the fusion and MDTB atlases, left-lateralized cognitive regions—such as D1 (spatial working memory) and S1 (linguistic processing) in the fusion atlas, and regions 5 (divided attention) and 7 (narrative processing) in the MDTB atlas—showed stronger age-related effects. In contrast, in the resting-state atlas, the corresponding dorsal attention and default mode networks exhibited age effects that were more bilaterally distributed across hemispheres.

Across all parcellations, we also observed sex differences, with males exhibiting overall higher age-related coefficients than females (Table 1). These differences were most pronounced in the right D2 (mean absolute difference in $\beta_{age} = 0.20$), left M3 (mean absolute difference in $\beta_{age} = 0.20$), and right S2 (mean absolute difference in $\beta_{age} = 0.20$) in the fusion atlas and region 1 (left hand press) in the MDTB atlas (mean absolute difference in $\beta_{age} = 0.20$) (p. 8, lines 234-248).

Discussion:

The three cerebellar functional atlases converged on a consistent developmental organization, characterized by flatter trajectories in sensorimotor regions and steeper trajectories in higher-order cognitive regions. At finer spatial scales, however, modest discrepancies emerged. The MDTB and functional fusion atlases showed highly consistent developmental profiles, including stronger age-related effects in left-lateralized cognitive regions associated with linguistic, attentional, and working memory processes. In contrast, the 7-network resting-state atlas exhibited more bilaterally distributed age effects in dorsal attention and default mode networks, likely reflecting its relatively coarse parcellation and bilateral parcel distribution, thereby limiting sensitivity to lateralized effects. Parcellation-dependent differences were also evident in sex-related effects, with sex differences being most pronounced in the fusion atlas, likely reflecting its finer-grained, multimodally-defined functional boundaries. Together, these findings indicate that finer-grained inferences regarding lateralization and sex differences might depend on atlas resolution and functional specificity. However, the consistency in sensorimotor and cognitive developmental patterns across different parcellation frameworks suggests that these patterns represent a robust and biologically grounded axis of cerebellar developmental organization⁶⁸ (p. 20-21, lines 497-511).

5. Although the authors employed nested 10-fold cross-validation for model selection and evaluation, the final behavioral prediction analyses were based on a single round of 10-fold CV using approximately 128 participants. Given the small sample size and potential variability of fold assignments, relying on a single split may yield unstable or optimistic estimates of predictive performance. A more rigorous approach would involve repeated cross-validation or external

validation on an independent dataset to ensure the robustness and generalizability of the reported brain–behavior associations.

We thank the Reviewer for raising this important point regarding the stability of cross-validated prediction estimates. We revised our analysis to implement repeated nested cross-validation, which provides a substantially more robust estimate of generalization performance. Specifically, we maintained the original inner 10-fold cross-validation for hyperparameter selection (Ridge, Lasso, Elastic Net), but replaced the outer evaluation step with a 10×10 repeated 10-fold cross-validation procedure. This approach evaluates each model across 100 unique train–test partitions, thereby reducing dependence on any single fold assignment and stabilizing performance estimates. For each behavior and feature set (cortex, cerebellum, combined), we computed predictive accuracy (R^2) within each outer fold and repeat, and the final reported values reflect the mean and standard error across all 100 paired resamples. Statistical comparisons between feature sets were conducted using Wilcoxon signed-rank tests applied to the paired R^2 values obtained across the repeated outer folds. The results remained consistent, with the cerebellum outperforming cerebral and combined cerebello-cerebral models in the prediction of social behavior, and now also language comprehension. We have revised the Methods, Results, and Figure 4E and Table 2 to reflect this change:

Revised Figure 4E with repeated cross-validation.

Methods:

Model selection and evaluation were performed using the same nested cross-validation procedure described in the previous section (Association with normative trajectories of the cerebral cortex), except here the inner 10-fold cross-validation was used solely for hyperparameter tuning, whereas model performance was quantified using a repeated 10-fold

outer cross-validation procedure. By repeating the outer cross-validation 10 times, we obtained 100 independent train–test partitions, from which predictive accuracy (R^2) was averaged to provide a stable estimate of how well cerebellar, cerebral, and combined normative features predicted behavioral outcomes (p. 33, lines 876-883).

Results:

We compared how well normative z-scores from cerebellum-only, cerebral-only, and combined cerebello-cerebral models predicted individual variations in language comprehension, social behavior, and reading abilities, using ElasticNet predictive performance (R^2) averaged across a repeated 10-fold cross-validation procedure. Cerebellum-only models performed comparably to cerebral-only and combined cerebello-cerebral models for reading abilities (pairwise Wilcoxon signed-rank tests, all $p > .05$, two-sided; **Table 2**). However, the cerebellum outperformed other models for language comprehension (cerebellum vs. cerebrum: $W = 623$, $N = 100$, $p < .001$; cerebellum vs. combined: $W = 1,590$, $N = 100$, $p < .001$; all two-sided) and social behavior measured via the Social Responsiveness Scale³¹ (SRS; higher scores indicating higher autistic traits) (cerebellum vs. cerebrum: $W = 42$, $N = 100$, $p < .001$; cerebellum vs. combined: $W = 387$, $N = 100$, $p < .001$; all two-sided) (**Figure 4E and Table 2**). Importantly, the cerebellar predictive performance was the highest for social behavior compared to other behaviors. This suggests a pronounced association between individual differences in normative cerebellar development and (atypical) social behavior across the developmental span from childhood to adulthood, over and above development of the cerebral cortex (p. 14-15, lines 366-381).

6. The paper also suffers from imprecise descriptions, inconsistent figure scaling, incomplete or unclear captions, and missing units in figures and tables, which together reduce the clarity, readability, and professionalism of the manuscript. The overall writing could be more formal, precise, and logically organized.

We thank the Reviewer for this feedback and agree that clarity and precision in presentation are essential. In the revised manuscript, we have carefully reviewed the text, figures, and tables to improve formal tone, consistency, and readability, and we have revised several descriptions and captions to enhance clarity and completeness.

Regarding missing units, the only instances where we found units absent were empty cells in Table 2, where values were intentionally omitted to avoid redundant repetition across rows. In the revised version, we have now explicitly filled all cells with units to prevent ambiguity and improve clarity. With respect to figure scaling, we z-scored regression coefficients in Figure 4D to provide a consistent scale across subplots and facilitate direct comparison. In Figures 1B and 2A-C (which are the remaining ones with inconsistent scaling), scaling choices were made intentionally to preserve interpretability of the data. Figures 1B and 2A–C depict absolute volumetric growth trajectories across parcels with substantially different sizes. Because parcel volumes vary widely (e.g., ranging from approximately 4 to 46 cm³ in the functional atlases),

rescaling these plots to a common y-axis would compress or exaggerate trajectories and obscure meaningful differences in growth relative to each parcel's baseline size. We therefore retained parcel-specific scaling in these figures to accurately convey developmental change within each region. To improve transparency, we have clarified these scaling choices explicitly in the corresponding figure captions:

Figure 1 and 2 captions:

Trajectories are shown on parcel-specific y-axis scales to preserve differences in absolute volume and to visualize growth relative to each lobule's baseline size (p. 7, lines 211-213 and p. 9, 258-260).

Reviewer #2 (Remarks to the Author):

Summary

The study presents several noteworthy findings, particularly the detailed characterization of cerebellar growth trajectories across childhood and adolescence and their relationship to cognitive development. The use of both lobular and functional parcellations provides convergent evidence supporting the cerebellum's dual motor and cognitive roles, which aligns well with and extends previous work (e.g., Stoodley & Schmahmann, 2009; Guell et al., 2018). The inclusion of two independent cohorts and robust statistical modeling strengthens the generalizability of the results. Comparisons of maturation with the cerebral cortex are very interesting.

The work is significant for developmental neuroscience and related fields, offering novel insights into how cerebellar maturation supports emerging cognitive abilities. The methodology is sound and consistent with current standards in neuroimaging and developmental modeling. The analyses are clearly described, and the level of methodological detail appears sufficient for replication.

The data support the authors' main conclusions, and the interpretation is generally well balanced. While some aspects (e.g., corpus medullare analyses) would benefit from clarification, these do not undermine the overall findings.

We thank the Reviewer for their positive evaluation of our manuscript!

1°) Anterior vs. posterior trajectories

(Line 196) The comparison between anterior and posterior lobes could be expanded. Within the posterior lobe, lobule VIIIA is known to contribute to motor functions despite its posterior location (Guell et al., 2018). This is reflected in Figures 2A–B, where the functional parcellation shows a mirror of anterior/motor regions near lobule VIIIA. Supplementary Table 1A shows notably lower beta values for VIIIA than for other posterior lobules. This suggests distinct developmental trajectories between motor and cognitive territories, which may be a more meaningful distinction than an anterior–posterior division.

We thank the Reviewer for this helpful observation and agree that a motor–cognitive distinction provides a more functionally meaningful framework than a strict anterior–posterior division. In response, we have revised the manuscript to consistently refer to “motor” versus “higher-order cognitive” territories when discussing results derived from functional parcellations. This change explicitly acknowledges that certain posterior lobules, including lobule VIIIA, serve motor functions, consistent with prior functional mapping work (e.g., Buckner et al., 2011; Guell et al., 2018; Magielse et al., 2025).

For analyses based on anatomical lobular boundaries (Figure 1 and the Results section titled “*Posterior cerebellar lobules exhibit steeper trajectories than anterior lobules from infancy to adulthood*”), we retain anterior–posterior terminology, as lobular subdivisions are anatomically

defined and not uniformly aligned with functional distinctions (King et al., 2019). However, we explicitly highlight the distinct developmental profile of lobule VIIIA within this section, noting its comparatively lower age-related coefficients relative to other posterior lobules, consistent with its motor functional specialization:

A notable exception was the bilateral posterior VIIIA, primarily involved in motor processing^{37,47}, which demonstrated flatter trajectories compared to other posterior lobules (left: $\beta_{age} = 0.01$; 95% CI = [-0.48 0.48]; right: $\beta_{age} = 0.05$; 95% CI = [-0.50 0.61]) (p. 6, lines 199-201).

2°) Corpus medullare analyses

The inclusion of the corpus medullare warrants further justification. This region comprises total cerebellar white matter, including deep nuclei, and its overall volume may not be functionally interpretable. It is unclear whether any quality control was performed for this structure, which is difficult to segment accurately. Given its absence from the functional atlas used elsewhere and the large sex differences reported in Supplementary Table 1, the results may reflect segmentation variability. Unless reliability can be demonstrated, these analyses should be removed or clearly qualified.

We thank the Reviewer for raising this important point and agree that analysis of the corpus medullare presents several challenges, including limited functional interpretability and potential segmentation uncertainty. Although the corpus medullare is automatically provided as a region of interest by the ACAPULCO segmentation pipeline, and steep developmental trajectories for this structure have been reported in prior work (e.g., Gaiser et al., 2024), we agree that its volumetric trajectory is difficult to interpret and may be more susceptible to segmentation-related variability. In light of these considerations, we have removed reports of the corpus medullare from the main results. Specifically, we replaced the corpus medullare with right lobule IV in Figure 1B, which also exhibits a steep developmental trajectory and is more readily interpretable within the framework of cerebellar functional organization. In addition, we replaced the report of large sex differences previously attributed to the corpus medullare with more modest sex differences observed in left lobules IX and V, which followed in the original results. Findings for the corpus medullare are now presented only in the Supplementary Materials for completeness and validation purposes, given that this region is included in the ACAPULCO segmentation.

Results:

The largest sex differences were observed in the left IX (mean absolute difference in $\beta_{age} = 0.38$) and left V (mean absolute difference in $\beta_{age} = 0.34$) (p. 6, lines 203-205).

Revised Figure 1 without corpus medullare trajectory.

3°) Interpretation of SRS results

The Social Responsiveness Scale (SRS) primarily assesses social functioning in autism spectrum conditions. As the HCP-D dataset excludes individuals with neuro developmental diagnoses, the range of SRS scores is likely limited. Reporting the distribution of SRS scores and commenting on their interpretability in this restricted sample would clarify how meaningful these associations are.

We thank the Reviewer for this important point. The distributions of all behavioral measures, including the SRS, are provided in **Supplementary Figure 14** (see also the figure below). We now more explicitly reference these distributions in the main text. As noted by the Reviewer, the HCP-D sample excludes individuals with diagnosed neurodevelopmental conditions, and SRS scores therefore reflect subclinical variation in social functioning within a typically developing population. We have added text to the Discussion clarifying that SRS was treated as a continuous measure indexing individual differences in social abilities rather than as a clinical indicator, and that no diagnostic or clinical inferences were made. While the restricted range of scores may limit effect sizes, we note that meaningful brain–behavior associations can still be detected within healthy population normative samples, and such associations may be particularly informative for understanding how variation in cerebellar development relates to social functioning across the typical range.

Supplementary Figure 14. Distribution of participants' scores of the selected behavioral tasks in the Human Connectome Project Development dataset ($N = 457$; 5-21 years; participants without behavioral scores were removed).

Discussion:

Importantly, in the present typically developing sample, SRS scores reflect subclinical variation in social functioning rather than diagnostic status. We here consider SRS scores as indexing individual differences in social abilities across the typical development spectrum and interpret them as a continuous behavioral measure rather than a clinical indicator. While subclinical score ranges may attenuate observed effects, the observation of significant cerebellar-behavior associations within this constrained distribution suggests that cerebellar developmental trajectories are particularly relevant for individual differences in social functioning even within typical development (p. 22, lines 549-556).

4°) Relation to Gaiser et al. (2024)

There is some overlap with Gaiser et al. (2024, *Nature Communications*), which also used normative modeling in pediatric data and examined SRS associations. Expanding the discussion to highlight similarities, differences, and unique contributions of the current work would improve context and novelty.

We thank the Reviewer for this important comment. In the revised Introduction and Discussion, we now explicitly contextualize our findings relative to Gaiser et al. (2024) by highlighting both points of convergence and key distinctions. Similar to that study, we observe steeper developmental trajectories in higher-order cognitive cerebellar regions relative to sensorimotor territories and associations between cerebellar development and individual differences in social functioning. We now explicitly acknowledge these similarities.

At the same time, we clarify how the present work extends beyond prior normative modeling efforts. Specifically, we emphasize that our study provides a comprehensive reference of cerebellar lobular growth spanning infancy through early adulthood, systematically validates developmental patterns across multiple cerebellar parcellation frameworks, and links cerebellar growth trajectories to concurrent structural maturation of the cerebral cortex. We further highlight that domain-specific cerebellar developmental trajectories show socio-linguistic behavioral associations that are comparable to or stronger than those observed in the cerebral cortex. We conclude by framing these findings as positioning cerebellar maturation as a central organizing axis of behavioral development and distributed cerebello–cerebral network formation across childhood and adolescence.

Introduction:

To investigate cerebellar development and how it relates to that of the cerebral cortex and behavioral outcomes, the present study describes normative models of cerebellar volumetric growth from infancy to young adulthood (p. 5, lines 151-153).

Together, these findings provide novel understanding of how cerebellar development aligns with cerebral maturation and supports emerging higher cognitive abilities, with potential relevance for identifying biomarkers of atypical neurodevelopment (p. 5-6, lines 173-176).

Discussion:

The consistent sensorimotor-higher cognitive differences in growth patterns across both lobular and functional parcellations mirror known patterns of cerebellar evolutionary expansion^{64,65} and are in line with previous cerebellar developmental research^{28-30,66}. Specifically, our findings align with recent normative modeling studies of cerebellar development, which likewise reported steeper volumetric growth trajectories in higher-order cognitive regions relative to sensorimotor territories^{28,29}. However, the present study extends this literature in several important ways. First, we establish a comprehensive reference of cerebellar lobular growth spanning the full

developmental period from infancy through early adulthood. Second, we systematically validate these developmental patterns across multiple cerebellar parcellation frameworks, allowing us to distinguish robust organizational principles from parcellation-dependent effects. Third, we explicitly link cerebellar growth trajectories to concurrent structural maturation of the cerebral cortex, providing insight into coordinated cerebello–cerebral development. Finally, we show that normative cerebellar growth trajectories, particularly in posterior higher cognitive regions, exhibit domain-specific associations with socio-linguistic behavior that are comparable to or stronger than those observed in the cerebral cortex. Together, these findings advance the growing literature on cerebellar development by positioning cerebellar maturation as a central organizing axis for behavioral development and the formation of distributed cerebello-cerebral networks across childhood and adolescence (p. 19, lines 461-478).

5°) The abstract states that behavioral results are presented for the entire sample (ages 1–21 years), which is inaccurate. Behavioral data were only available for the HCP-D cohort, and this limitation should be clearly specified to avoid confusion.

We thank the Reviewer for noting this inconsistency. We have revised the abstract to clarify that behavioral associations were performed only in children and adolescents (for whom behavioral measures were available) to avoid confusion:

Using lobular and functional cerebellar parcellations, we comprehensively characterize typical cerebellar development from infancy and examine how it aligns with cerebral development and behavioral outcomes from childhood to adulthood (lines 53-55).

Reviewer #2 (Remarks on code availability):

The code is available and could be usable.

Reviewer #3 (Remarks to the Author):

Manoli et al. used normative modelling to map cerebellar volumetric growth in n=751 infants and young adults. Further, the team investigated how cerebellar growth is linked to cerebral cortical development and cognitive and behavioral outcomes. An advance of this analysis is that both structural (lobular) and functional parcellations were used to evaluate the data, which provides a finer-grained analysis that accounts for the relatively large size of some cerebellar lobules and the known medial-to-lateral functional organization of the cerebellum.

The age effects are consistent with known behavioral / brain developmental trajectories, with sensorimotor regions maturing earlier than those associated with higher cognitive functions. This has been previously established in the cerebellum by other developmental studies (e.g. Tiemeier et al 2010). The link to behavior is a nice addition to this study, although it is only possible in a more restricted age range (5 yrs and up) due to available data.

In the analysis of the co-development of the cerebellar parcels and the cerebral cortical parcels, though a major concern is that the cerebellum was parcellated functionally and the cerebral cortical parcellation was a structural parcellation. This is particularly important in the comparison between how well cerebellar vs cerebral cortical regions correlated with behavioral scores and should be addressed.

Overall, this study adds to the expanding literature regarding cerebellar development and the relationship between cerebellar development and emerging behaviors. The available growth curves will be updated with new data as it becomes available, which is a nice resource for the field. The additional analyses investigating co-development of the cerebellum and cerebral cortex and linking cerebellar development to behaviors take full advantage of the data available and lead to a more comprehensive picture. That said, there are some concerns about the analyses as they stand, which are outlined below.

We thank the Reviewer for the positive evaluation of our work! We have responded to all concerns raised below.

Questions / comments:

Why did the authors exclude the BCP data from birth-age 1?

Thank you for this question. BCP data from birth to one year of age were excluded due to severe limitations in cerebellar segmentation quality using the ACAPULCO pipeline. In this age range, T1-weighted images provide insufficient tissue contrast for reliable cerebellar lobular segmentation. Although T2-weighted images offer relatively improved contrast in early infancy, this image modality cannot be accurately processed by the ACAPULCO algorithm. In many cases, segmentations were incomplete, mislocalized, or extended outside the cerebellum altogether. Visual inspection of the raw images confirmed that cerebellar lobular boundaries were

often not discernible at this developmental stage, making both automated and manual segmentation unreliable and highly assumption-dependent. Including these data would therefore introduce substantial noise and bias into the normative models. This is in part due to the fact that, to date, there is no reliable pipeline for cerebellar segmentation from infancy to adulthood, which we already note in the Discussion:

This is partly attributable to the fact that, to date, there is no reliable cerebellar segmentation method that covers the entire developmental range from the perinatal period to adulthood. Different algorithms (e.g., SUI^{78,79}, ACAPULCO⁴³) perform variably across age ranges, requiring laborious manual correction (p. 22, lines 565-568).

To ensure transparency, we have added example scans illustrating the extent of segmentation failure in this age range below:

ACAPULCO segmentation in younger BCP participants.

Please provide the final sample size per age bin in the main text Methods, after exclusion, so that it is easier to understand how much data is represented at different ages. Were there similar numbers of males / females in each bin?

We thank the Reviewer for this suggestion. In response, we have added a new Methods figure (“Participants” section; Figure 5) illustrating the age distribution of participants in the final sample after all exclusions. This figure displays the number of participants per age bin and stratifies counts by sex, with overlapping bars indicating the number of male and female participants within each bin.

Figure 5. Age distribution of participants by sex in the final sample. Overlapping bars indicate the number of male and female participants within each age bin.

Data is cross-sectional and not longitudinal, which should be emphasized in the limitations; these are not brains that are followed over time, so we do not know how an individual might track over development. This is an extremely common issue in human longitudinal data, so it is not a “problem” per se, but should be mentioned in the Discussion.

We thank the Reviewer for this important remark and agree that the cross-sectional nature of the data should be explicitly emphasized as a limitation. In the revised Discussion, we now expand

our discussion of this point to clearly state that developmental trajectories are inferred at the cohort level and do not reflect within-individual change over time. While this limitation is common in large-scale developmental neuroimaging studies, longitudinal data will be required to determine how individual brains track relative to normative trajectories over time and to identify timing-sensitive developmental effects:

Lastly, in this study we focused on establishing normative cerebellar growth trajectories in a typically developing cross-sectional cohort. As such, the reported developmental trajectories reflect age-related variation at the cohort level and cannot determine how a given individual brain changes across development. While this design does not allow the inference of timing-sensitive, causal, or mechanistic processes underlying cerebellar–cerebral maturation, these normative models nevertheless provide a necessary reference framework for future work aimed at elucidating such mechanisms. Longitudinal studies will be required to determine when deviations from normative trajectories first emerge, how cerebellar and cerebral maturation dynamically influence one another within individuals, and whether specific periods of cerebellar vulnerability or plasticity can be identified. Future research should therefore validate and extend our models in longitudinal or clinical populations to assess individual variability over time⁸⁹ or in or in neurodevelopmental and psychiatric conditions. For example, comparing cerebellar subregional volumes for ASD patients to our reference model could reveal its utility in ASD diagnosis. In the same vein, it can be assessed whether cerebellar or cerebello-cerebral deviations from the reference model predict patients’ behavioral outcomes. At later stages, prognostic brain scans in young children may be used to predict increased risk for several disorders or behavioral deficits. Such applications could enable early detection of cerebellar abnormalities and inform targeted interventions. The ability to quantify individual phenotypes relative to age-matched normative reference ranges represents a crucial step toward precision medicine approaches in pediatric neurology and psychiatry, particularly for conditions where cerebellar involvement is suspected or established²⁵ (p. 23-24, lines 590-610).

The ACAPULCO pediatric parcellation was trained on the ABIDE dataset, which includes ages 5 and up – how confident are the authors in using this algorithm for the 1-5 year olds? While the authors manually checked and corrected the images, this may be a challenge to do on a large scale for other studies. Were more corrections needed for the younger age ranges?

We thank the Reviewer for raising this important question, which directly concerns both the validity and scalability of our approach. We acknowledge that ACAPULCO was trained on data from older children (≥ 5 years). However, in the present study, all segmentations underwent rigorous multi-reviewer quality control, and manual corrections were applied where necessary to ensure anatomical plausibility. Importantly, as shown in Supplementary Figure 2, manual corrections generally resulted in relatively modest changes to lobular volumes, suggesting that the primary developmental patterns reported here are not driven by segmentation artifacts (see also Supplementary Figure 3 for ACAPULCO's performance in younger children).

Supplementary Figure 2. Example lobular segmentations performed with the ACAPULCO algorithm. Left: Example ACAPULCO segmentation, with a native space lobular mask overlaid on a participant's T1-weighted image (sagittal and coronal views). Right: Example of cerebellar overinclusion (top; circled in red) and underinclusion (bottom; circled in red). For both types of segmentation errors, slice-wise manual correction of the mask was performed by tracing cerebellar fissures and folia (see **Methods: Cerebellar parcellation**). In the case of overinclusion, the ACAPULCO mask included parts of the cerebral cortex, non-brain infratentorial tissue, and/or lower skull and neck tissue. In the case of underinclusion, the ACAPULCO mask often missed parts of the cerebellar grey matter. Both errors were most common around postero-lateral cerebellar regions.

Supplementary Figure 3. Example lobular segmentations performed with the ACAPULCO algorithm in the BCP dataset (see **Methods: Cerebellar parcellation**). Native space ACAPULCO segmentation masks are overlaid on participants’ T1-weighted images (sagittal and coronal views) in example timepoints (1, 3, and 5 years) across the BCP age range (1-5 years).

Segmentation quality was, to an extent, age-dependent. In the youngest age ranges, particularly between 1 and 2 years of age, ACAPULCO performance was regularly suboptimal, and manual correction was required to obtain more accurate cerebellar segmentations (see figure below for an example “suboptimal” scan in a 1-year-old participant). Manual correction was less frequent in older children aged over 2 years.

Example cerebellar underinclusions (circled in red) in a 1-year-old participant.

While thorough visual inspection and manual correction (as necessary) improved anatomical accuracy, it was highly labor intensive (approximately 3 hours per scan), and we therefore agree that such manual curation is not a scalable solution for large population-based or clinical normative modeling studies. In the current study, this additional effort was undertaken to establish a proof-of-principle normative cerebello–cerebral developmental framework and to maximize confidence in the reported brain–behavior associations. For future large-scale applications and clinical translation, there is a clear need for automated cerebellar segmentation tools specifically trained on, and validated in, early childhood (0–5 years). Our findings further suggest that imaging quality itself plays a critical role, as particularly segmentation in the 0–1 year BCP data proved unreliable across methods, likely due to insufficient cerebellar tissue contrast. We discuss these methodological considerations and their implications for future normative modeling efforts in the revised manuscript:

Discussion:

First, our sample size is on the smaller end of population-wide normative modeling endeavors. This is partly attributable to the fact that, to date, there is no reliable cerebellar segmentation method that covers the entire developmental spectrum from the perinatal period to adulthood. Different algorithms (e.g., SUI^{79,80}, ACAPULCO⁴³) perform variably across age ranges, requiring laborious manual correction. Second, minimal age overlap between HCP-D and BCP datasets means age effects may be confounded with site effects, potentially limiting generalizability. Despite these constraints, our findings use careful quality control and manual correction of cerebellar segmentations to provide a comprehensive framework for cerebellar development from the first year of life through adulthood, bridging previous studies that examined infant³⁰ and child/adolescent populations^{28,29} separately. Our publicly available normative models can be further validated and expanded as new data becomes available^{25,28}. Future efforts should develop reliable cerebellar functional parcellation methods spanning the entire developmental trajectory, which will enable more robust normative modeling and accurate assessment of individual deviations across all developmental stages (p. 22-23, lines 564-577).

The MDTB task battery parcellations are based on data from young adults, who are outside the age range of this study – though the authors cite other studies showing similarities in parcellations across ages (particularly in older children and adolescents), are these functional parcellations appropriate for the younger age range in this study? Would a voxel-based analysis provide a less constrained approach?

We thank the Reviewer for raising this important point. We agree that applying adult-based functional parcellations to developing populations introduces relatively strong priors regarding the spatial organization of function, and we now discuss this limitation more explicitly in the revised manuscript.

To mitigate this concern, we adopted a multi-atlas strategy rather than relying on a single functional framework, allowing us to assess the robustness of developmental patterns across different parcellation schemes. This is also in line with prior frameworks emphasizing convergence across different parcellations as a criterion for biological plausibility (e.g., Eickhoff et al., 2018). Importantly, we restricted the application of adult-based functional cerebellar parcellations to participants aged 5–21 years. This age threshold provided a natural boundary between the BCP and HCP-D datasets and is supported by evidence that cerebellar functional topography undergoes rapid reorganization primarily between birth and early childhood (0–5 years), after which large-scale functional organization becomes increasingly similar to that observed in adults (e.g., Lyu et al., 2025; Sun et al., 2025; Yates et al., 2021). In addition, task-based studies suggest that cerebellar functional territories supporting specific cognitive domains, such as Theory of Mind, show substantial overlap between children and adults from early childhood onward (e.g., Manoli et al., 2025). While this does not imply complete developmental stability of functional boundaries, it supports the—albeit cautious—use of adult-based parcellations in school-aged children and adolescents.

We also appreciate the Reviewer’s suggestion to consider voxel-wise analyses as a less constrained alternative. We considered this approach carefully; however, voxel-wise normative models would still ultimately require interpretation within a functional framework to yield biologically meaningful conclusions. Interpreting voxel-wise effects by reference to adult-based functional atlases would therefore raise similar concerns regarding developmental appropriateness. In this context, using established parcellations provides an explicit and transparent reference frame for interpretation, and comparing results across multiple parcellations allows us to assess convergence and robustness. Notably, prior normative modeling work in pediatric samples, including Gaiser et al. (2024), similarly employed the MDTB cerebellar atlas rather than a voxel-wise approach, enabling direct comparison and extension of existing findings.

Although age-specific functional atlases for different developmental stages in early childhood have begun to emerge (e.g., Lyu et al., 2025), integrating different age-specific parcellations within a single normative modeling framework introduces additional challenges in reconciling developmental effects across atlas definitions. We therefore view the development of age-resolved cerebellar functional atlases spanning multiple developmental stages as an important direction for future work, and we now explicitly highlight this need in the Discussion:

A further limitation is our application of adult-based functional cerebellar parcellations to developing populations. Even though children and adults may recruit similar cerebellar regions for cognitive functions as early as around the third year of life (e.g., Theory of Mind³²), functional boundaries in the cerebellum likely vary as a function of age³². Here, we restricted our

functional atlas modeling to children and adolescents between 5-21 years. From this age onward, functional similarity to adult cerebellar organization is considered sufficient to support the use of adult-based parcellations^{48,49,81}. However, this approach assumes that functional organization in the cerebellum remains stable across development, which likely does not fully capture age-related changes in cerebellar functional topography⁸¹. Future research should create age-specific functional atlases of the cerebellum across different functional tasks and developmental stages, allowing for more developmentally appropriate parcellation schemes that could reveal age-dependent spatial changes in cerebellar functional organization (p. 23, lines 578-589).

“Our results show that posterior cerebellar regions involved in higher association processes undergo more rapid developmental growth than anterior sensorimotor regions” – the authors should emphasize that these findings are in this age group and that they are missing the 0-1 year old data, which is an age range when there is rapid development in sensorimotor networks and behaviors.

Thank you for bringing this to our attention. We have now rephrased this sentence in the Introduction to include the precise age range that these findings apply to:

Our results show that posterior cerebellar regions involved in higher cognitive processes undergo more rapid developmental growth than sensorimotor regions from approximately one year of age through adulthood (p. 5, lines 167-169).

Major concern: Why was the DK atlas used for the cerebral cortical parcellation? The authors parcellated the cerebellum functionally but the cerebral cortex structurally. There are cortical parcellations that reflect e.g. functional networks (such as the Yeo / Buckner maps) that might better reflect co-growth between the cerebellum and cerebral cortex. The Yeo / Buckner parcellations would allow a “like to like” comparison between the cerebral cortex and cerebellum. This is particularly important given the authors’ interpretation of the co-development of “functionally similar” cerebral cortical and cerebellar regions.

We thank the Reviewer for this valuable methodological suggestion and agree that aligning cerebellar and cerebral parcellations within a functional framework is important for interpretable cerebello–cerebral comparisons. Accordingly, we revised our analyses to incorporate the Yeo et al. 17-network cerebral parcellation, which provides a level of functional granularity comparable to the cerebellar functional fusion atlas (Nettekoven et al., 2024). We fit normative developmental models for each cerebral resting-state network and examined their maturational coupling with cerebellar functional regions. These analyses were conducted using the cerebellar functional fusion atlas to maintain comparability with the original results based on the Desikan–Killiany atlas.

Notably, the findings were highly consistent with those obtained using the anatomical cerebral parcellation. In cerebello–cerebral coupling analyses, cerebellar social regions (S2 and S3) showed positive associations with default mode network components in the Yeo et al. 17-network atlas (e.g., network 15). Motor cerebellar regions exhibited expected network specificity, with M2 selectively coupled to sensorimotor and visual networks (e.g., networks 4 and 1), while M3 showed negative associations with default mode networks (e.g., network 16). The multi-demand cerebellar region D2, associated with retrieval processes, demonstrated strong coupling with higher-order association networks, including salience and limbic networks (e.g., networks 7 and 9). Together, these results demonstrate the robustness of cerebello–cerebral associations across both anatomical and functional cerebral parcellation schemes. We have accordingly revised the Methods and added a detailed presentation of these analyses in Supplementary Figure 12.

Supplementary Figure 12. Associations of cerebellar (fusion atlas) and cerebral (Yeo et al. 17-network atlas) growth trajectories. **A.** Functional fusion atlas (adapted with permission from Nettekoven et al.⁶). **B.** Mean effect of age on the Yeo et al. 17-network atlas (top) and example normative trajectory for right

network 17 (default mode; bottom). Bold lines represent the mean trajectories for each sex. Shaded areas represent the 95.5% confidence interval. **C.** Comparison of Ridge, Lasso, and ElasticNet regularization models based on average R^2 scores (10-fold cross-validation). The Lasso model marginally outperformed the other two and was selected (mean $R^2 = 0.34$). Error bars represent the standard error of the mean. **D.** Top: Significant cerebello-cerebral associations (10,000 permutations, FDR corrected at $q = .05$). A global $\alpha = 0.1$ was selected for associations based on a multioutput Lasso model predicting all cerebral parcels simultaneously and aligns with the distribution of parcel-wise optimal α values, where over 80% of cerebral parcels individually favored $\alpha = 0.1$. Bottom: FDR-corrected weights for cerebellar parcels with the largest number of significant associations in the fusion atlas, projected on the Yeo 17-network atlas. Abbreviations: Coef. = coefficient; E. Net = ElasticNet; VIS = visual network; SMN = somatomotor network; DAN = dorsal attention network; SAL = salience network; LIM = limbic network; FPN = frontoparietal network; TPN = temporoparietal network; DMN = default mode network; M = male; F = female; L = left; R = right.

Why not perform voxel-based analyses? Particularly in the younger ages, performing analyses using parcellations established in adults might not harness the full power of the dataset and the parcellations institute constraints on the data. This approach should be better rationalized, especially in applying the parcellations to a different age group. What is the advantage of this over data-driven approaches which then can be interpreted in the context of the functional parcellations?

We thank the Reviewer for this thoughtful question and agree that voxel-based analyses represent a more data-driven alternative. We ultimately opted for a parcellation-based approach for several reasons related to interpretability, comparability, and developmental inference.

First, voxel-wise normative models would still require interpretation within a functional framework to yield biologically meaningful conclusions. Interpreting voxel-wise developmental effects typically involves mapping them onto functional atlases, many of which are likewise derived from adult data. As such, voxel-based analyses would not fully circumvent the issue of developmental appropriateness and would raise similar interpretive challenges when referenced to adult-defined functional boundaries.

Second, our primary goal was to characterize developmentally meaningful patterns of cerebellar organization and cerebello-cerebral coupling that are interpretable at the systems level and comparable across studies. Using established parcellations provides an explicit and transparent reference frame for interpretation, and examining convergence across multiple parcellation schemes allows us to distinguish robust developmental features from parcellation-specific effects. This strategy also facilitates direct comparison with prior normative modeling work in pediatric samples, including Gaiser et al. (2024), which similarly employed the MDTB cerebellar

atlas rather than a voxel-wise approach in a developmental sample with a similar age range (6-17 years).

Finally, although age-specific functional atlases for different developmental stages in early childhood have begun to emerge (e.g., Lyu et al., 2025), integrating different age-resolved parcellations within a single normative modeling framework introduces additional challenges in reconciling developmental effects across atlas definitions and age ranges. We therefore view the development of continuous, age-resolved cerebellar functional atlases as an important future direction. We explicitly discuss the limitations of our chosen approach in the Discussion:

A further limitation is our application of adult-based functional cerebellar parcellations to developing populations. Even though children and adults may recruit similar cerebellar regions for cognitive functions as early as around the third year of life (e.g., Theory of Mind³²), functional boundaries in the cerebellum likely vary as a function of age³². Here, we restricted our functional atlas modeling to children and adolescents between 5-21 years. From this age onward, functional similarity to adult cerebellar organization is considered sufficient to support the use of adult-based parcellations^{48,49,81}. However, this approach assumes that functional organization in the cerebellum remains stable across development, which likely does not fully capture age-related changes in cerebellar functional topography⁸¹. Future research should create age-specific functional atlases of the cerebellum across different functional tasks and developmental stages, allowing for more developmentally appropriate parcellation schemes that could reveal age-dependent spatial changes in cerebellar functional organization (p. 23, lines 578-589).

The analysis showing how cortical vs cerebellar maturation is associated with individual behavioral scores similarly suffers from the fact that the cortical parcellations are structural and the cerebellar parcellations are functional. Task based cortical parcellation would be a more appropriate comparison here.

We thank the Reviewer for this comment and agree that using functionally defined cerebral parcellations provides a more appropriate comparison with cerebellar functional parcellations in brain-behavior analyses. In response, we repeated the behavioral prediction analyses using the Yeo et al. 17-network functional cerebral atlas, consistent with the cerebello-cerebral coupling analyses reported elsewhere in the manuscript.

Using the same analytic framework, we evaluated cerebellar-only, cerebral-only, and combined cerebello-cerebral models and assessed predictive performance using repeated 10-fold cross-validation, as suggested by Reviewer 1, to improve robustness. The results remained consistent with those reported in the main text: cerebellar models continued to outperform cerebral-only and combined cerebello-cerebral models in predicting social behavior, reinforcing

the cerebellum's unique contribution to socio-behavioral development. These additional analyses are presented in Supplementary Figure 16.

Supplementary Figure 16. Cerebellar (fusion), cerebral (Yeo 17-networks), and combined cerebello-cerebral parcel z -scores were compared for how well they predicted socio-linguistic behaviors using ElasticNet regularization. Left: Mean Lasso, Ridge, and ElasticNet model performance (R^2) across cerebellar, cerebral, and combined models for each behavior, averaged over cross-validation. Error bars represent the standard error of the mean. Right: ElasticNet model performance (R) for cerebellum-only, cerebral-only, and combined model across language, reading abilities and social behavior, averaged over repeated cross-validation. Error bars represent the standard error of the mean. Abbreviations: E. Net = ElasticNet.

p. 20 “The pronounced influence of age on higher association posterior cerebellar regions may suggest that the cognitive functions these areas support undergo substantial refinement during childhood”. The authors should reference extensive developmental literature on the emergence and maturation of different behaviors over the course of childhood and adolescence to support this statement in a more specific way. Further, they can evaluate this statement with the data used in the current study.

We thank the Reviewer for this helpful suggestion. In the revised manuscript, we have strengthened this statement by explicitly grounding it in the developmental neuroimaging literature linking brain maturation to the emergence and refinement of higher-order cognitive and social functions across childhood and adolescence. Specifically, we now cite multiple developmental studies demonstrating that higher cognitive functions such as language (e.g.,

Skeide & Friederici, 2025), executive control (e.g., Satterthwaite et al., 2013), and social cognition (e.g., Richardson et al., 2018) emerge alongside ongoing structural and functional brain development, including within the cerebellum (e.g., Manoli et al., 2025):

Discussion:

This is in line with prior studies showing a relationship between brain development and the early-life emergence of higher cognitive functions such as language^{72,73}, executive control^{74,75}, and social cognition^{76,77} including within the cerebellum³² (p. 21, lines 513-515).

While we agree that evaluating this interpretation empirically is an important goal, we note that the cross-sectional design of the current study limits our ability to directly test within-individual developmental refinement of cognitive functions. We therefore did not introduce additional analyses beyond those already reported. Instead, we frame our interpretation cautiously as being consistent with converging evidence from prior developmental neuroimaging work, rather than as a direct test within the present dataset. We believe this approach appropriately contextualizes our findings while respecting the methodological constraints of the study.

Minor comments / suggestions:

- **Another cerebellar developmental study to add for interpretation / discussion in typically-developing children and adolescents is Moore et al. 2017 Dev Cog Neuroscience. This study showed differential relationships between cerebellar grey matter volume and behavioral scores during development.**

Thanks for suggesting this interesting paper! We referenced these findings in the Discussion of our revised manuscript (p. 22, line 545).

- **Instead of referring to functional parcels by e.g. “D1” in the main text, a brief description would be helpful for readers e.g. “spatial working memory”; this is done in some places but is not consistent.**

Thank you for bringing this to our attention. We have added a brief description of the functional role of each parcel based on its definition in the functional fusion atlas (Nettekoven et al., 2024) throughout the manuscript.

- **Were the behavioral scores from the HCP the age-adjusted scores or the raw scores?**

The behavioral scores were age-adjusted. We have now clarified this in the Methods (“Behavioral tasks”):

All scores were age-adjusted (p. 26, line 676).

Response to Reviewers - NCOMMS-25-76551-A

We thank the Editor and the Reviewers for their thoughtful re-evaluation of our work, their appreciation of the revisions made, and their constructive suggestions for further improving the manuscript. We believe that the Reviewers' comments have substantially enhanced the clarity, rigor, and interpretability of the study. In response, we have implemented the following key revisions.

First, we have moved the analyses based on the Yeo et al. 17-network atlas added in the first revision from the Supplementary Information to the main text. This change facilitates “like-to-like” comparisons between the cerebellar functional fusion atlas and a functional cerebral parcellation for cerebello-cerebral co-maturation and behavioral analyses. As noted in the previous revision, the results are consistent with the original analyses using the gyral Desikan-Killiany atlas.

Second, we quantified interhemispheric similarity for cerebello-cerebral associations based on the Desikan-Killiany atlas that were averaged for visualization purposes in the original Figure 3D (now Supplementary Figure 11D). The median Spearman correlation across homologous parcel pairs was $r = 0.71$ (interquartile range: 0.61-0.81), indicating a high degree of correspondence between hemispheric patterns and supporting that hemispheric averaging does not mask meaningful associations. We have included these results in the manuscript.

Third, we corrected the cross-validation inference procedure, by aggregating performance metrics within each of the 10 full outer cross-validation repeats, yielding $N = 10$ repeat-level estimates used for statistical inference. This correction was applied consistently to both the Yeo and Desikan-Killiany analyses and yielded results consistent with the original findings, namely that cerebellar features show superior predictive performance compared with cerebral-only or combined cerebello-cerebral features for social behavior.

In addition, we made several writing and reporting clarifications. These include replacing “infancy” with “late infancy” or “from age one year” throughout the manuscript, revising the use of the term “outcomes” in the title and text to avoid implying longitudinal prediction, adding a limitation noting the relatively limited amount of behavioral data in the 5-7-year age range, and providing version-pinned requirement files specifying package dependencies in the GitHub repository.

Detailed responses to each Reviewer comment are provided below. In the revised manuscript, all changes are highlighted in yellow and are also summarized in the response document.

Reviewer #1 (Remarks to the Author):

The authors have substantially revised the manuscript and provided a detailed rebuttal. Several of my major methodological concerns have been well addressed, particularly regarding the choice of cortical parcellation, the improved robustness of behavioral predictions, and the discussion of atlas-dependent discrepancies.

There are still several concerns that should be addressed to ensure internal consistency and statistical accuracy.

We thank the Reviewer for their constructive feedback and helpful further suggestions.

1. About Hemispheric averaging in DK heatmap, I appreciate that the authors have now provided hemisphere-specific results in Supplementary Figure 11 and clarified that the averaging in Figure 3D was performed solely for visualization purposes. However, the decision to present averaged data in the main figure creates a logical tension with the revised Discussion. The authors now argue that discrepancies between atlases (e.g., Fusion vs. Resting-State) are partly driven by their differing sensitivity to lateralization, specifically, that the Fusion atlas captures left-lateralized cognitive effects while the Resting-State atlas is more bilateral. If lateralization is a key factor explaining the results (or the differences between atlases), averaging the hemispheres in the primary figure risks obscuring the very biological properties the manuscript now highlights. I strongly recommend moving the hemisphere-specific heatmap (currently Supplementary Figure 11) to the main text to replace the averaged plot. Alternatively, if the averaged plot is retained, please quantify the correlation between left and right hemispheric patterns to statistically justify that averaging does not mask significant asymmetries.

Thank you for this thoughtful comment. In line with your and Reviewer 3's suggestion, we have swapped the cerebellar-cerebral association figures, moving the Desikan-Killiany-based associations to the Supplementary Information (now Supplementary Figure 11) and bringing the Yeo et al. 17-network atlas-based associations into the main text (now Figure 3). Importantly, for the Yeo 17-network atlas we do not average across hemispheres, and our primary conclusions are therefore drawn from the hemisphere-specific results shown in the main figure (Figure 3D).

To further address the concern that hemispheric averaging might obscure biologically meaningful lateralization effects in the Desikan-Killiany associations, we additionally quantified interhemispheric similarity for the analyses in which hemispheres were averaged for visualization purposes. Specifically, we computed Spearman correlations between homologous left-right parcel pairs for the associations between the functional fusion cerebellar atlas and the cerebral Desikan-Killiany atlas (Supplementary Figure 11). The median Spearman correlation across parcel pairs was $r = 0.71$ (interquartile range: 0.61-0.81). We performed the same analysis for associations between the lobular cerebellar atlas and the cerebral Desikan-Killiany atlas (Supplementary Figure 13), yielding a median correlation of $r = 0.70$ (interquartile range: 0.60-0.80).

These results indicate a high degree of interhemispheric correspondence, suggesting that hemispheric averaging in these specific visualizations does not mask substantial asymmetries. We have added these quantitative results to the captions of Supplementary Figures 11D and 13D to make this justification explicit.

2. The rebuttal correctly describes the implementation of a 10x10 repeated outer cross-validation, resulting in 100 total fold partitions. The authors state that they performed Wilcoxon signed-rank tests using N=100. However, in a repeated cross-validation framework, the 100 folds are not statistically independent samples; the same data points are reused across the 10 repeats. Treating 100 as the sample size for the significance test inflates the degrees of freedom and may yield overly optimistic p-values. Please clarify the sampling unit used for inference. A more conservative and standard approach would be to aggregate performance metrics within each of the 10 full cross-validation repeats (resulting in N=10) or to use a corrected t-test designed for repeated cross-validation. Please ensure the reported N, degrees of freedom, and p-values in the text and Table 2 reflect the dependency structure of the repeated folds.

Thank you for pointing this out. We agree that treating all 100 folds from the repeated outer cross-validation as independent samples is inappropriate due to dependency across repeats, and we have corrected our inference procedure accordingly. Specifically, the outer cross-validation was repeated 10 times, yielding 10 sets of 10 train-test partitions. Within each repeat, predictive performance (R^2) was first averaged across the 10 folds to produce a single performance estimate per repeat. Statistical inference was then performed on these repeat-level performance estimates ($N = 10$), thereby respecting the dependency structure of the repeated cross-validation. Performance differences between cerebellar-only, cerebral-only, and combined feature sets were evaluated using pairwise Wilcoxon signed-rank tests across the 10 cross-validation repeats. All reported sample sizes (N), test statistics, and p-values have been updated to reflect this corrected procedure.

The corrected results are reported in the Results section (*Growth of cerebellar higher cognitive regions corresponds to socio-linguistic behavioral development*, p. 13), in Figure 4E and Table 2 (relative contributions of cerebellum vs. Yeo 17-network atlas), and in Supplementary Figure 16 and Table 10 (relative contributions of cerebellum vs. Desikan-Killiany atlas). Importantly, the main conclusions remain unchanged. Across analyses, cerebellar features consistently demonstrated superior predictive performance compared with cerebrum-only or combined cerebello-cerebral feature sets for social behavior. In the primary analyses, comparing the cerebellum with the Yeo 17-network atlas (Figure 4E), the cerebellum showed superior predictive power across all socio-linguistic behavioral measures compared to cerebrum-only models.

Results:

“Across all three behavioral domains, cerebellum-only models consistently outperformed cerebral-only models (pairwise Wilcoxon signed-rank tests across 10 cross-validation repeats, all two-sided $p < .05$). For reading abilities and language comprehension, cerebellum-only models performed comparably to combined cerebello-cerebral models (all two-sided $p > .05$), indicating no additional benefit of incorporating cerebral features. In contrast, for social behavior as measured by the Social Responsiveness Scale³¹ (SRS; higher scores indicate greater autistic traits), cerebellum-only models significantly outperformed both cerebral-only and combined models (cerebellum vs. cerebrum: $W = 0.0$, $N = 10$, $p = .002$; cerebellum vs. combined: $W = 0.0$, $N = 10$, $p = .002$; all two-sided) Figure 4E and Table 2 (relative contributions of cerebellum vs. Yeo 17-network atlas), and Supplementary Figure 16 and Table 10 (relative contributions of cerebellum vs. Desikan-Killiany atlas). Importantly, the main conclusions remain unchanged. Across analyses, cerebellar features consistently demonstrated superior predictive performance compared with cerebrum-only or combined cerebello-cerebral feature sets for social behavior. In the primary analyses comparing the cerebellum with the Yeo 17-network atlas (Figure 4E), the cerebellum showed superior predictive power across all socio-linguistic behavioral measures compared to cerebrum-only models.” (p. 13, lines 384-392)

Figure 4E. Cerebellar (fusion), cerebral (Yeo et al. 17-networks), and combined cerebello-cerebral parcel z-scores were compared for how well they predicted socio-linguistic behaviors using ElasticNet regularization. Left: Mean Lasso, Ridge, and ElasticNet model performance (R^2) across cerebellar, cerebral, and combined models for each behavior, averaged over cross-validation. Error bars represent the standard error of the mean. Right: ElasticNet model performance (R^2) for cerebellum-only, cerebral-only, and combined model across language, reading abilities and social behavior, averaged over repeated cross-

validation. Error bars represent the standard error of the mean. The functional fusion atlas was adapted with permission from Nettekoven et al.⁶.

Table 2. Comparison of ElasticNet performance for cerebellum vs. other models

Behavior	Comparison	$M_{Cerebellum}$	$SD_{Cerebellum}$	M_{Other}	SD_{Other}	W	N	p
Reading	Cerebellum vs. Cerebral Cortex	-0.01	0.01	-0.02	0.01	7.0	10	.030*
	Cerebellum vs. Combined	-0.01	0.01	-0.01	0.01	11.0	10	.105
Language Comprehension	Cerebellum vs. Cerebral Cortex	0.05	0.01	-0.01	0.02	0.0	10	.002*
	Cerebellum vs. Combined	0.05	0.01	0.06	0.01	72.0	10	.210
Social Behavior (SRS)	Cerebellum vs. Cerebral Cortex	0.25	0.02	0.14	0.02	0.0	10	.002*
	Cerebellum vs. Combined	0.25	0.02	0.22	0.03	0.0	10	.002*

* = $p < .05$, two-sided.

Supplementary Figure 16. Cerebellar (fusion), cerebral (DK), and combined cerebello-cerebral parcel z-scores were compared for how well they predicted socio-linguistic behaviors using ElasticNet regularization. Left: Mean Lasso, Ridge, and ElasticNet model performance (R^2) across cerebellar, cerebral, and combined models for each behavior, averaged over cross-validation. Error bars represent the standard error of the mean. Right: ElasticNet model performance (R) for cerebellum-only, cerebral-only, and combined model across language, reading abilities and social behavior, averaged over repeated cross-validation. Error bars represent the standard error of the mean. Abbreviations: E. Net = ElasticNet.

Supplementary Table 10. Comparison of ElasticNet performance for cerebellum vs. other models (Desikan-Killiany cerebral atlas)

Behavior	Comparison	$M_{Cerebellum}$	$SD_{Cerebellum}$	M_{Other}	SD_{Other}	W	N	p
Reading	Cerebellum vs. Cerebral Cortex	-0.01	0.01	0.01	0.02	0.0	10	.006*
	Cerebellum vs. Combined	-0.01	0.01	0.01	0.02	0.0	10	.002*
Language Comprehension	Cerebellum vs. Cerebral Cortex	0.05	0.01	-0.01	0.01	0.0	10	.002*
	Cerebellum vs. Combined	0.05	0.01	0.04	0.01	1.0	10	.002*
Social Behavior (SRS)	Cerebellum vs. Cerebral Cortex	0.25	0.02	0.05	0.03	0.0	10	.002*
	Cerebellum vs. Combined	0.25	0.02	0.18	0.04	0.0	10	.002*

* = $p < .05$, two-sided

The revised cross-validation and inference procedure is described in detail in the Methods section (*Association with behavioral performance*):

“By repeating the outer cross-validation 10 times, we obtained 10 sets of 10 train-test partitions. Predictive accuracy (R^2) was first averaged across folds within each repeat, yielding one performance estimate per repeat, and then summarized across repeats to provide a stable estimate of how well cerebellar, cerebral, and combined normative features predicted behavioral scores.” (p. 33, lines 898-902)

“Performance differences between cerebellar-only, cerebral-only, and combined feature sets were evaluated using pairwise Wilcoxon signed-rank tests on repeat-level performance estimates across the ten cross-validation repeats ($N = 10$).” (p. 33, lines 904-907)

3. There appears to be a discrepancy in the reported regularization parameters. The revised figure caption for Supplementary Figure 12 states that "A global $\alpha = 0.1$ was selected" for the Lasso model. However, the caption text provided in the Supplementary Figure 14 describes a "global $\alpha = 1000$ " for the same analysis. Please review the manuscript and supplementary captions to resolve this discrepancy and ensure the correct parameter value or mathematical symbol is reported.

We apologize for the confusion. We had inadvertently misidentified the selected model in Supplementary Figure 13D, referring to it as Lasso rather than Ridge. This has now been corrected, and Supplementary Figure 13D correctly labels the chosen model as Ridge.

The reported alpha parameters are however correct, as they correspond to different analyses: the fusion cerebellum-Yeo cortex analysis (originally Supplementary Figure 12, now Figure 3 in the main text) and the anatomical cerebellum-Desikan-Killiany cortex analysis (Supplementary Figure 13).

4. I note that data from birth to 1 year of age were excluded due to segmentation limitations. While this exclusion is methodologically sound given the imaging constraints, the manuscript continues to use the broad term "infancy" in the Abstract and overarching descriptions. Since "infancy" typically encompasses the first year of life, using the term without qualification may be imprecise. Please consider refining the terminology (e.g., "from late infancy" or "from 1 year of age") to accurately reflect the specific developmental window covered by the normative models.

Thank you for this helpful suggestion. We agree that the unqualified use of the term infancy could be imprecise given that data from birth to 1 year of age were excluded. We have therefore replaced the term “infancy” with “late infancy” or “from age one year” throughout the manuscript, accurately reflecting the developmental window covered by the normative models.

Reviewer #1 (Remarks on code availability):

While the repository provides a well-structured analysis workflow for cerebellar normative modeling, it would benefit from a version-pinned environment file (e.g., environment.yml or requirements.txt) to improve reproducibility and ease of use.

Thank you for this suggestion. We have now added version-pinned requirement files to the project's GitHub repository, specifying the Python package dependencies used for the analyses.

Reviewer #2 (Remarks to the Author):

The authors have addressed all my concerns. In my opinion, this paper is suitable for publication.

Reviewer #2 (Remarks on code availability):

The code is usable

We thank the Reviewer for their contributions that have greatly improved our manuscript and are happy they consider our work suitable for publication.

Reviewer #3 (Remarks to the Author):

The revised manuscript has improved in several key areas noted by the reviewers, including a more critical and comprehensive assessment of the limitations of the study and better couching of the study in the context of the prior literature (e.g. recent work by Gaiser et al.).

We thank the Reviewer for their positive feedback and for their favorable appraisal of the revisions made in response to their previous suggestions.

That said, two of the three reviewers noted that comparing structural cortical parcellations with functional cerebellar parcellations is problematic. The authors conducted the recommended cortical functional parcellation analyses, but these have been pushed into supplement - it seems that that should be the primary analysis for cerebello-cortical coupling, or at the very least for assessing the relative contributions of the cerebral cortex and the cerebellum to behavioral scores. In the abstract, it is stated that “cerebellar and cerebral areas with similar functional roles demonstrate coordinated maturation”, but this is better tested with similar parcellations in both structures. Further, the comparison between cerebellar predictors of socio-linguistic behaviors and cortical predictors is using functional cerebellar parcels and structural cortical parcels, with the functional cortical parcels in the supplement; in this case, the cerebellum still shows additional predictive power over the cerebral cortex, but it is less marked than when the cortex is parcellated anatomically. These are important considerations.

We thank the Reviewer for raising this important point. In response, we have revised the manuscript so that the functional cerebellar-functional cortical analyses based on the Yeo 17-network atlas are now presented in the main text (Figures 3 and 4E). Associations with the Desikan-Killiany atlas are moved to Supplementary Figures 11 and 16. The Yeo 17-network analyses now form the primary framework for assessing cerebello-cortical coupling and for comparing the relative contributions of the cerebellum and cerebral cortex to behavioral prediction.

Importantly, the results and conclusions remain consistent after this reorganization. Using matched functional parcellations, we continue to observe domain-specific associations between cerebellar and cerebral networks, supporting the claim that cerebellar and cerebral regions with similar functional roles demonstrate coordinated maturation. As we describe in the Results (*Cerebellar and cerebral subregions demonstrate domain-specific maturational coupling*):

“The most prominent associations within functional domains of the fusion atlas⁶ (i.e., socio-linguistic, multi-demand, motor) exemplify this domain-specific pattern. In the socio-linguistic domain, right S2 (corresponding to social processing) and S3 (corresponding to resting-state activation) were primarily associated with default-mode network regions (e.g., networks 16-17). In the multi-demand domain, right D2 (corresponding to working memory), was associated with attention, memory, and executive control regions (e.g., dorsal attention network 5, salience networks 7-8, and limbic network 9). Right M2

(corresponding to mouth movements) in the motor domain was strongly correlated with sensorimotor networks (e.g., visual network 2 and somatomotor network 4). Lastly, right M3 (corresponding to lower limb movements) was strongly anticorrelated with default-mode networks 15-17, but notably also with somato-motor network 3. No significant associations were found for the action domain at the FDR threshold (Figure 3D).” (Results, p. 10-11, lines 287-297)

2

Figure 3. Associations of cerebellar (fusion atlas) and cerebral (Yeo et al. 17-network atlas) growth trajectories. **A.** Functional fusion atlas (adapted with permission from Nettekoven et al.⁶). **B.** Mean effect of age on the Yeo et al. 17-network atlas (top) and example normative trajectory for right network 17 (default mode; bottom). Bold lines represent the mean trajectories for each sex. Shaded areas represent the 95.5% confidence interval. **C.** Comparison of Ridge, Lasso, and ElasticNet regularization models based on average R^2 scores (10-fold cross-validation). The Lasso model marginally outperformed the other two and was

selected (mean $R^2 = 0.34$). Error bars represent the standard error of the mean. **D.** Top: Significant cerebello-cerebral associations (10,000 permutations, FDR corrected at $q = .05$). A global $\alpha = 0.1$ was selected for associations based on a multi-output Lasso model predicting all cerebral parcels simultaneously and aligns with the distribution of parcel-wise optimal α values, where over 80% of cerebral parcels individually favored $\alpha = 0.1$. Bottom: FDR-corrected weights for cerebellar parcels with the largest number of significant associations in the fusion atlas, projected on the Yeo et al. 17-network atlas. Abbreviations: Coef. = coefficient; E. Net = ElasticNet; VIS = visual network; SMN = somatomotor network; DAN = dorsal attention network; SAL = salience network; LIM = limbic network; FPN = frontoparietal network; TPN = temporoparietal network; DMN = default mode network; M = male; F = female; L = left; R = right.

Moreover, in the behavioral prediction analyses, cerebellar features consistently demonstrated superior predictive performance compared with cerebrum-only or combined cerebello-cerebral feature sets for social behavior. Additionally, in the analyses comparing the cerebellum with the Yeo 17-network cortical atlas (Figure 4E), the cerebellum showed superior predictive power across all socio-linguistic behavioral measures relative to cerebrum-only models.

“Across all three behavioral domains, cerebellum-only models consistently outperformed cerebral-only models (pairwise Wilcoxon signed-rank tests across 10 cross-validation repeats, all two-sided $p < .05$). For reading abilities and language comprehension, cerebellum-only models performed comparably to combined cerebello-cerebral models (all two-sided $p > .05$), indicating no additional benefit of incorporating cerebral features. In contrast, for social behavior as measured by the Social Responsiveness Scale³¹ (SRS; higher scores indicate greater autistic traits), cerebellum-only models significantly outperformed both cerebral-only and combined models (cerebellum vs. cerebrum: $W = 0.0$, $N = 10$, $p = .002$; cerebellum vs. combined: $W = 0.0$, $N = 10$, $p = .002$; all two-sided) (Figure 4E and Table 2).” (Results, p. 14, lines 384-392)

Figure 4E. Cerebellar (fusion), cerebral (Yeo et al. 17-networks), and combined cerebello-cerebral parcel z-scores were compared for how well they predicted socio-linguistic behaviors using ElasticNet regularization. Left: Mean Lasso, Ridge, and ElasticNet model performance (R^2) across cerebellar, cerebral, and combined models for each behavior, averaged over cross-validation. Error bars represent the standard error of the mean. Right: ElasticNet model performance (R^2) for cerebellum-only, cerebral-only, and combined model across language, reading abilities and social behavior, averaged over repeated cross-validation. Error bars represent the standard error of the mean. The functional fusion atlas was adapted with permission from Nettekoven et al.⁶.

Table 2. Comparison of ElasticNet performance for cerebellum vs. other models

Behavior	Comparison	$M_{Cerebellum}$	$SD_{Cerebellum}$	M_{Other}	SD_{Other}	W	N	p
Reading	Cerebellum vs. Cerebral Cortex	-0.01	0.01	-0.02	0.01	7.0	10	.030*
	Cerebellum vs. Combined	-0.01	0.01	-0.01	0.01	11.0	10	.105
Language Comprehension	Cerebellum vs. Cerebral Cortex	0.05	0.01	-0.01	0.02	0.0	10	.002*
	Cerebellum vs. Combined	0.05	0.01	0.06	0.01	72.0	10	.210
Social Behavior (SRS)	Cerebellum vs. Cerebral Cortex	0.25	0.02	0.14	0.02	0.0	10	.002*
	Cerebellum vs. Combined	0.25	0.02	0.22	0.03	0.0	10	.002*

* = $p < .05$, two-sided.

We agree with the Reviewer that when the functional Yeo 17-network parcellation is used, the difference between cerebellar and cerebral predictive performance is less pronounced than when the cortex is parcellated anatomically. However, the overall pattern of results remains consistent, and these differences remain statistically significant (Table 2). The reduced magnitude of the differences when using functional cortical parcellations likely reflects a combination of factors, including methodological differences between anatomical and functional parcellations as well as conceptual considerations related to structure-function correspondence. Notably, when both cerebellar and cortical features are defined functionally, their predictive performance becomes more similar, which may account for the attenuated differences observed and highlights the challenges inherent in directly comparing structure-based and function-based representations, in line with the Reviewer's argument. Disentangling the relative contributions of these methodological and biological factors would require additional analyses beyond the scope of the present study, and we therefore restrict our interpretation to the observed patterns.

Other things to be addressed:

- In the abstract, the phrasing “growth ... aligns with cerebral development and behavioral outcomes” implies a longitudinal analysis, where change over time reflects later behavior. More appropriate phrasing would be something like “how it corresponds with cerebral cortical development and brain-behavior relationships during childhood through adulthood”. Similarly, it is recommended that “outcomes” in the title be removed, given that this is not a longitudinal cohort - the cerebellar growth index is associated with socio-linguistic behavior at the same timepoint, rather than behavioral outcomes. This may sound fussy, but it these edits are important to accurately reflect the analyses.

Thank you for this careful and important clarification. We agree that the original phrasing could be interpreted as implying a longitudinal analysis and have thus revised the language to more accurately reflect the cross-sectional nature of the data.

Specifically, in the Abstract we now state:

“Using both lobular and functional cerebellar parcellations, we characterize typical cerebellar development from late infancy and its relationship to cerebral development and behavioral performance in childhood through adulthood.” (lines 53-55)

This wording avoids implying temporal precedence or outcome prediction and instead emphasizes contemporaneous brain-behavior relationships.

In addition, we have revised the title by replacing “behavioral outcomes” with “behavior”, and throughout the manuscript we have replaced the term “outcomes” with more precise alternatives such as “behavioral performance”, “behavioral scores”, or “behavioral development”, as appropriate. These changes ensure that the terminology accurately reflects the analyses conducted.

- Given that the authors have excluded data from birth-age 1, the data do not include the infancy period, which ends at 1 year of age. This should be clarified throughout the manuscript. There also is a limited amount of data before age 7, which should be noted in the limitations, and the functional and cerebro-cerebellar analyses are only conducted in children aged 5+. This is a limitation of the dataset and not by design, but again the edits are needed to accurately report the study.

Thank you for highlighting this important point. We agree that the age range covered by the dataset should be described more precisely. In response, and also following Reviewer 1's suggestion, we have revised the manuscript throughout to refer to “late infancy” or “from age one year” rather than infancy, consistent with the exclusion of data from birth to 1 year of age.

We have also added a clarification in the limitations section noting that there is a relatively limited amount of data between ages 5 and 7.

“Moreover, behavioral data in the 5-7 year age range were limited, which may have reduced sensitivity to detect age-specific effects. Future research should develop age-specific functional atlases of the cerebellum across different tasks and developmental stages, ideally with larger samples, to enable developmentally appropriate parcellation schemes that could reveal age-dependent spatial changes in cerebellar functional organization.” **(Discussion, p. 23, lines 603-607)**